# Statistical and functional convergence of common and rare genetic influences on autism at chromosome 16p

Daniel J. Weiner [1,2] ✉, Emi Ling [1,3], Serkan Erdin [4,5,6], Derek J. C. Tai[4,5,6], Rachita Yadav[4,5,6], Jakob Grove [7,8,9,10], Jack M. Fu[1,4,5,6], Ajay Nadig[1,2], Caitlin E. Carey[1,6,11], Nikolas Baya[1], Jonas Bybjerg-Grauholm [10,12], iPSYCH Consortium*, ASD Working Group of the Psychiatric Genomics Consortium*, ADHD Working Group of the Psychiatric Genomics Consortium*, Sabina Berretta[13,14], Evan Z. Macosko[1,15], Jonathan Sebat[16], Luke J. O'Connor [4], David M. Hougaard [10,12], Anders D. Børglum [7,8,10], Michael E. Talkowski [1,4,5,6], Steven A. McCarroll[1,3] and Elise B. Robinson [1,6,15] ✉

The canonical paradigm for converting genetic association to mechanism involves iteratively mapping individual associations to the proximal genes through which they act. In contrast, in the present study we demonstrate the feasibility of extracting biological insights from a very large region of the genome and leverage this strategy to study the genetic influences on autism. Using a new statistical approach, we identified the 33-Mb p-arm of chromosome 16 (16p) as harboring the greatest excess of autism's common polygenic influences. The region also includes the mechanistically cryptic and autism-associated 16p11.2 copy number variant. Analysis of RNA-sequencing data revealed that both the common polygenic influences within 16p and the 16p11.2 deletion were associated with decreased average gene expression across 16p. The transcriptional effects of the rare deletion and diffuse common variation were correlated at the level of individual genes and analysis of Hi-C data revealed patterns of chromatin contact that may explain this transcriptional convergence. These results reflect a new approach for extracting biological insight from genetic association data and suggest convergence of common and rare genetic influences on autism at 16p.

Genome-wide association studies (GWAS) have productively identified robust statistical associations between thousands of common genetic variants and traits[1]. However, most associations are noncoding, complicating efforts to identify the genes that mediate these associations[2,3]. A dominant approach is to fine-map associations to individual variants and then to their nearby target genes[4,5]. Although there are numerous examples of success[5,6], functional interpretation of individual genetic variants remains a critical bottleneck. Moreover, most complex trait heritability often does not reside within these individually significant associations, but is rather scattered across thousands of individually nonsignificant loci across the genome[7].

Autism spectrum disorder (ASD, autism) provides a compelling example for the need to jointly interpret many classes of genetic variation[8–14]. Although common polygenic variation is the largest genetic influence on autism at a population level, extracting biological insight from this predominantly noncoding signal is challenging[11,15]. Similarly,

de novo recurrent copy number variants (CNVs), which are strongly associated with autism, often encompass many genes with generally undefined downstream mechanisms[8,13,16–18]. For example, although deletion of the 0.7-Mb, 31-gene locus at chromosome 16p11.2 is one of the most common and largest single genetic influences on autism[8,19], exactly how the deletion increases the likelihood of autism diagnosis has remained undetermined despite considerable inquiry[20–24]. Thus, a critical open question is whether regional polygenic signals colocalize with recurrent large CNVs and whether this colocalization can highlight uncommonly relevant areas of the genome for autism diagnosis. In particular, given that both regions of common polygenic influence and recurrent large CNVs span many genes and influence chromatin structure and gene regulatory landscapes[25–28], large chromatin landscapes have the potential to unify analysis of regional polygenic and rare variations.

To examine polygenic influences arising from regions of the genome, including regions harboring autism-associated CNVs, we developed the stratified polygenic transmission disequilibrium test (S-pTDT), which extends the trio-based polygenic transmission disequilibrium test (pTDT) to genomic annotations. Using S-pTDT and 9,383 European ancestry autism trios, we performed an unbiased genome-wide search for excess over-transmission of autism's polygenic influences. Unexpectedly, the greatest excess is localized to the 33-Mb p-arm of chromosome 16 (16p), the region that includes the recurrent, autism-associated and mechanistically cryptic proximal 16p11.2 CNV. Further linking the 16p11.2 CNV with the broader p-arm of the chromosome, in vitro deletion of the 16p11.2 locus was associated with decreased average expression of neuronally expressed genes on chromosome 16p. Similarly, an increased autism polygenic score (PGS) constructed exclusively with 16p variants was associated with decreased average expression of cortically expressed genes on 16p across multiple cohorts. These transcriptional effects of the 16p11.2 deletion and 16p autism PGS were correlated at the level of individual genes on 16p, suggesting mechanistic convergence of common and rare variant influences on autism in the region. We observed chromatin contact patterns that we hypothesize explain this transcriptional convergence: uncommonly high within-16p chromatin contact in two independent Hi-C datasets and increased contact between the 16p11.2 locus and a distal region on 16p (Mb: 0–5.2) with convergent gene expression changes. Our results motivate a model of convergent common and rare genetic influences on autism at 16p and more broadly suggest that chromatin contact may facilitate coordinated genetic and transcriptional effects within very large regions of the genome.

## Results

### S-pTDT identifies over-transmission of autism PGS at 16p

Individuals with autism inherit more common polygenic influences on autism from their parents than expected by chance ('over-transmission')[12]. Using pTDT, we observe mean over-transmission of autism PGS within European ancestry trios from three different autism trio cohorts: the Psychiatric Genomics Consortium (PGC)[11] ($n = 4,335$ trios, 0.20 s.d. over-transmission, $P = 1.5 \times 10^{-37}$), the Simons Simplex Collection (SSC)[29] ($n = 1,851$ trios, 0.19 s.d. over-transmission, $P = 1.3 \times 10^{-17}$) and Simons Foundation Powering Autism Research (SPARK)[30] ($n = 3,197$ trios, 0.17 s.d. over-transmission, $P = 6.4 \times 10^{-21}$), all using a PGS generated from an external GWAS of the Danish iPSYCH consortium (19,870 individuals with autism and 39,078 controls; Supplementary Figs. 1 and 2). As biological insights from autism's common variant influences have been limited, we aimed to leverage the statistical power of pTDT to identify regions of the genome with excess common variant relevance in autism. We therefore developed S-pTDT, which estimates transmission in parent–child trios of PGS constructed from small sets of SNPs (Fig. 1a). Similar to pTDT, S-pTDT's within-family design prevents spurious association due to population stratification and many types of ascertainment bias[12].

We first asked whether S-pTDT could identify any regions of the genome with transmission of autism polygenic influence significantly over or under genome-wide expectation. To do this, we constructed stratified PGSs from adjacent blocks of SNPs, yielding 2,006 (often overlapping) partitions collectively covering the whole genome (median SNPs per partition: 3,000, median partition length: 11.7 Mb; Supplementary Fig. 3 and Methods). We then performed S-pTDT on each partition, first estimating transmission in the 5,048 trios from SSC + SPARK and then in the 4,335 trios from PGC. As expected given the robust over-transmission of the genome-wide autism PGS, most of the stratified partitions have a point estimate of over-transmission, and the degree of over-transmission increases with number of SNPs in the partition and size of the partition (Supplementary Fig. 4). To estimate the extent to which transmission of each region differed from expectation, we constructed a linear model regressing S-pTDT transmission on the number of SNPs in the partition and the length of the partition (Methods). This model yields a residual $z$-score for each partition, which estimates, in s.d., how much more or less transmission there is than there is expected relative to genome-wide patterns.

Transmission of regional polygenic influences on autism is correlated between SSC + SPARK and PGC trios ($r = 0.21$, $P < 1 \times 10^{-10}$; Fig. 1b and Supplementary Fig. 5), which indicates stability in the S-pTDT rankings despite each partition including on average only 0.3% of all PGS variants, the vast majority of partitions containing no autism GWAS loci, and phenotypic and genetic heterogeneity among the iPSYCH autism GWAS and autism trio cohorts. Partitions that include autism GWAS loci[11] are enriched among positive ($z$-score >0) S-pTDT scores in both SSC + SPARK and PGC trios: 29 of 46 (63%) partitions including an ASD GWAS locus are located in the top right quadrant, compared with expectation of 12 partitions ($P = 1.7 \times 10^{-8}$) (Fig. 1b, red points). This observation is consistent with the expectation that individuals with autism on average over-inherit alleles that increase the probability of autism diagnosis.

Unexpectedly, partitions with large S-pTDT $z$-scores cluster on 16p (approximately 0–33 Mb; Fig. 1b, blue points): of the 12 partitions with the largest average S-pTDT $z$-score across SSC + SPARK and PGC, 5 are on 16p, whereas the other 7 localize to autism GWAS loci. The three partitions that are nominally S-pTDT enriched in both datasets (S-pTDT $z$-score >1.96) collectively cover the entirety of 16p (Fig. 1c). Given that the highly over-transmitted regions span 16p, we constructed a single new 33-Mb partition spanning the p-arm and compared it with the 73 other, nonoverlapping, 33-Mb regions found in the human genome (Fig. 1d, inset). The excess over-transmission at 16p becomes even more apparent in this framing, with an S-pTDT $z$-score in SSC + SPARK of 3.37 (Fig. 1d). In contrast, the same common variants at 16p are not over-transmitted to 1,509 unaffected siblings in SSC (S-pTDT $z$-score = −0.06, $P = 0.95$; Supplementary Fig. 6).

Although 16p does not contain a genome-wide significant locus for autism (Supplementary Fig. 7), we nevertheless sought to determine whether the S-pTDT signal at 16p could be explained by one or a small number of common variant associations. We partitioned the 16p region into 25 adjacent blocks with low between-block linkage disequilibrium (median length: 1.3 Mb) and assessed the S-pTDT signal for each[31]. Consistent with the absence of a single driving locus in the region, the association signal was diffuse (Supplementary Fig. 8) and decayed gradually with successive removal of the most over-transmitted blocks (Fig. 1e and Methods). Restated, the S-pTDT association at 16p does not appear to be driven by a single coding or regulatory locus in the region, but exists more diffusely across the 33-Mb segment of genome.

We performed a number of additional analyses to further interrogate the S-pTDT finding at 16p. First, individuals with autism with a neurodevelopmental disorder-associated CNV on 16p (1.0% of individuals with autism in SSC + SPARK) did not drive the signal (Methods and Supplementary Fig. 9). Second, there was no association across the genome between S-pTDT ranking and either (1) presence of an

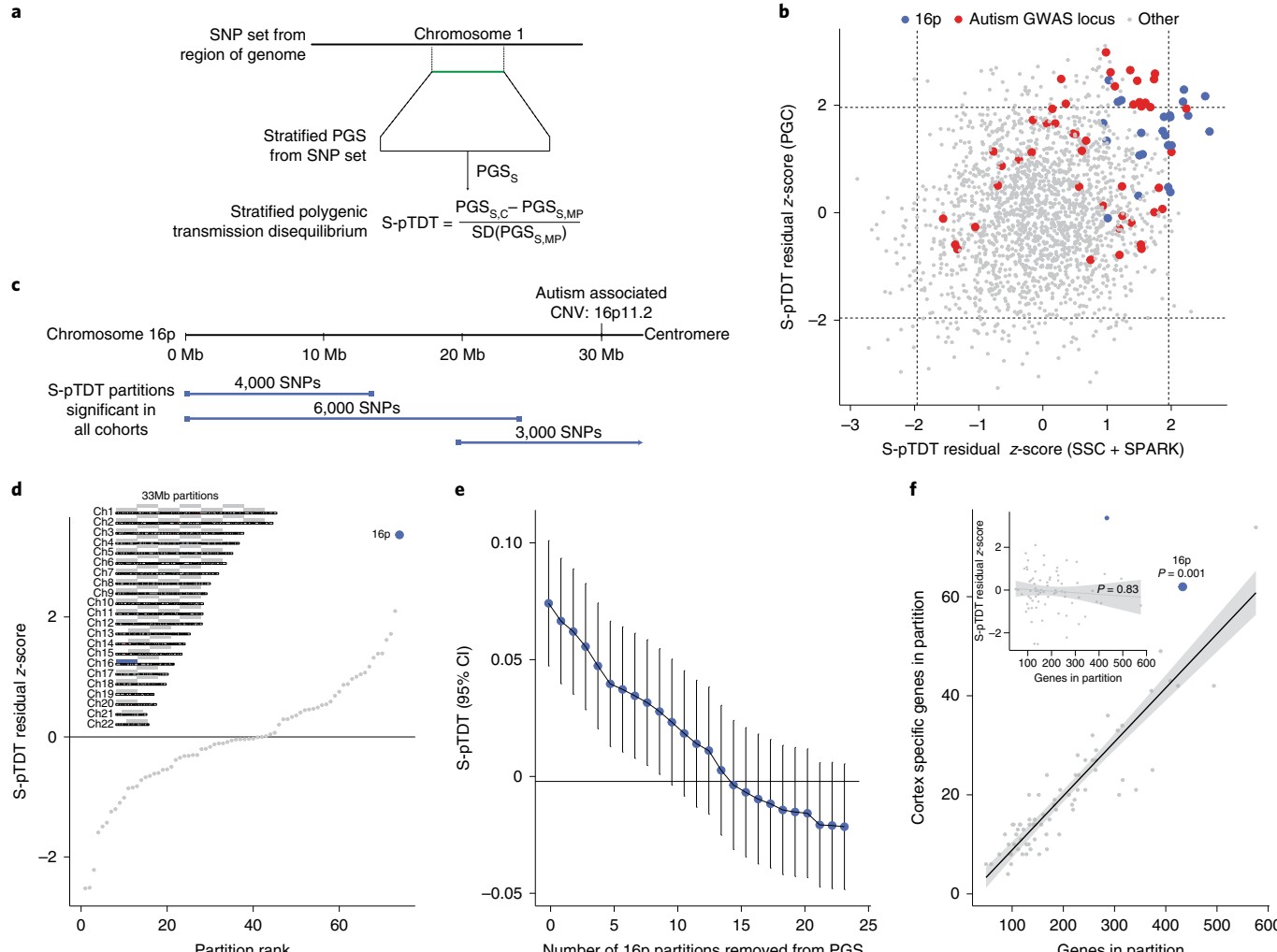

**Fig. 1 | S-pTDT identifies exceptional polygenic signal at chromosome 16p.**
**a**, S-pTDT estimates transmission of stratified PGSs from parents to their children. PGS$_S$ is a stratified PGS constructed from a continuous block of SNPs, denoted in green. The S-pTDT value for a parent–child trio is the normalized difference between the stratified PGS in the child and the average stratified PGS of their parents. **b**, Genome-wide S-pTDT analysis using European ancestry autism probands in the combined SSC + SPARK cohorts (x axis, $n = 5,048$ trios) and the PGC (y axis, $n = 4,335$ trios). Blue points are partitions on 16p and red points are partitions including an autism GWAS locus. Score weights are derived from an autism GWAS of the iPSYCH consortium. The axes are in units of S-pTDT residual z-scores (Methods) and dashed lines denote z-scores ±1.96. **c**, The three partitions that are nominally significant in both combined SSC + SPARK and PGC (blue bars) collectively span 16p. **d**, Autism S-pTDT analysis of the combined SSC + SPARK cohorts with stratified PGSs constructed from the 33-Mb partitions.

Inset: the 16p partition (blue bar) compared with 73 other 33-Mb partitions (gray bars) spanning the genome. **e**, SSC + SPARK autism S-pTDT ($n = 5,048$ trios) signal decaying gradually with successive removal of the most associated remaining linkage disequilibrium-independent block on 16p. The y axis is the S-pTDT estimate (±95% confidence interval (CI)) for transmission of a stratified PGS constructed from the union of the remaining blocks. **f**, Each point is shown as a 33-Mb partition as defined in **d**. Brain-specific genes are defined as genes in the top 10% of specific expression in cortex relative to nonbrain tissues in GTEx. The P value is calculated from the residual z-score in a linear model regressing the number of cortex-specific genes on the number of genes in the partition (two sided). Inset: the P value is from Pearson's correlation of number of genes in partition and S-pTDT residual z-score (two sided). For both the inset and the main panel, the trend line is a linear best fit and the shaded regions denote a 95% CI.

autism-associated CNV (Supplementary Fig. 10) or (2) segmental duplication content of the region (Supplementary Fig. 11). Third, we queried the specificity of the S-pTDT finding at 16p to autism by performing an analogous analysis using a cohort of 1,634 attention deficit hyperactivity disorder (ADHD) trios and an external iPSYCH ADHD GWAS; we did not replicate the finding in ADHD (Supplementary Fig. 12 and Methods). Finally, we did not see evidence that the autism S-pTDT signal at 16p extends to the q-arm of chromosome 16 (Supplementary Fig. 13). In summary, over-transmission of autism's polygenic influences at 16p is not driven by CNV carriers in our data, genomic structural features genome wide or as a crosstrait finding.

We analyzed the gene composition of 16p in relation to the 73 other 33-Mb control regions and asked whether gene density could explain the S-pTDT signal (Methods). With 433 genes, 16p is the third most gene-dense region (Fig. 1f). Furthermore, with 62 genes specifically expressed in the brain, 16p ranks second highest relative to the other 33-Mb control regions and has 37% more than predicted by the number of total genes—the greatest excess of any region ($P = 0.001$; Fig. 1f). In contrast, 16p does not have a significant excess of genes implicated in autism from exome associations studies ($P = 0.44$; Supplementary Fig. 14)[32]. Given that 16p exhibits polygenic over-transmission and is gene dense, we tested the hypothesis that polygenic over-transmission reflects gene density. Across all 74 33-Mb partitions, S-pTDT was not related to density of all genes ($r = 0.03$, $P = 0.83$; Fig. 1f inset), brain-specific genes ($r = 0.08$, $P = 0.48$), constrained genes ($r = -0.02$, $P = 0.85$) or genes associated with autism via exome sequencing

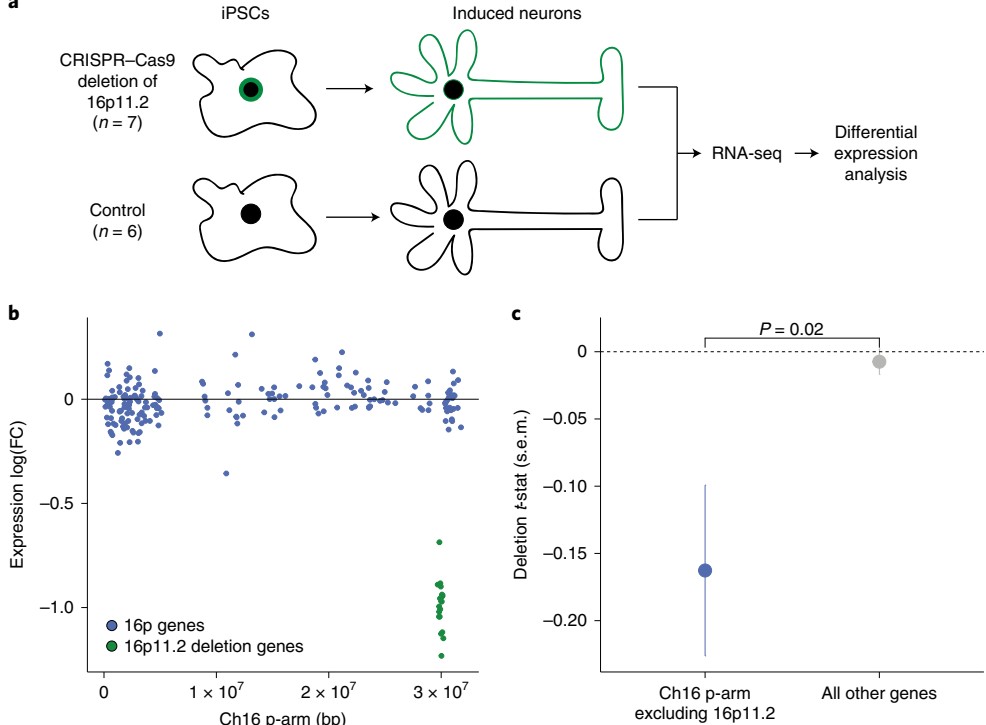

**Fig. 2 | In vitro deletion of 16p11.2 causes decreased average expression of neuronally expressed 16p genes. a**, Experimental design of 16p11.2 in vitro deletion resource. The iPSCs undergo CRISPR–Cas9-mediated deletion of the 16p11.2 CNV locus, differentiation into induced neurons and transcriptome profiling with RNA-seq (*n* = 7 biological replicates). Differential expression analysis compares these samples with controls (*n* = 6 biological replicates) without deletion of the locus. **b**, Differential expression of neuronally expressed genes on 16p (*n* = 200 genes) after deletion of the 16p11.2 locus (neuronally expressed is defined as above-median normalized expression level of genes

over all samples in analysis of induced neurons). FC, fold-change. Genes in the deletion region ±0.1 Mb are green, whereas all other genes on 16p are in blue. The *y* axis is the log₂(fold-change) per gene. **c**, The 16p11.2 deletion causing decreased expression of neuronally expressed genes on 16p (*n* = 200 genes), but not of all other neuronally expressed genes in the genome (*n* = 8,533 genes). Point estimates are of mean differential expression *t*-statistic for the group of genes ± s.e.m. The *P* value is from two-sided, two-sample Student's *t*-test comparing groups.

(*r* = 0.15, *P* = 0.21) (Supplementary Fig. 15). Moreover, we do not observe a trend for gene-dense regions having higher S-pTDT *z*-scores, for example, of the 13/74 partitions with >300 genes, the largest is 16p at *z* = 3.37 and the second largest at *z* = 1.32. We also did not observe a relationship between the S-pTDT signal and the density of fetal brain enhancers (*r* = 0.02, *P* = 0.88; Methods)[33]. This analysis suggests that although 16p is gene dense and enriched in brain-specific genes, these findings alone cannot explain a region's degree of polygenic over-transmission.

Finally, we sought to functionally characterize the genes on 16p. We did not observe an enrichment of genes differentially expressed between individuals with autism and controls in 16p (*P* > 0.68; Methods)[34]. We also performed gene ontology (GO) analysis of genes on 16p[35,36], but regional clustering of functionally related genes complicates interpretation (see Supplementary Table 1 for discussion). This challenge motivated us to pursue new functional approaches to characterize the genes on 16p.

### Deletion 16p11.2 causes decreased expression of 16p genes
Whereas the 16p11.2 CNV locus is 0.7 Mb, we observed S-pTDT signal across the entire p-arm of chromosome 16p. We hypothesized that the 16p11.2 deletion exerted effects on gene expression across 16p. A previous report with endogenous (nonengineered) 16p11.2 deletion lines noted differential expression effects extending up to 5 Mb from the 16p11.2 CNV[20]. We sought to extend the analysis to the entire p-arm using engineered 16p11.2 deletions on an isogenic background.

We analyzed a resource of clustered regularly interspaced short palindromic repeats (CRISPR)–Cas9-mediated heterozygous

deletion of the 16p11.2 locus in induced pluripotent stem cells (iPSCs; *n* = 7 biological replicates)[37]. These iPSCs were differentiated into NGN2-induced neurons, and differential expression analysis was performed on the deletion lines relative to control neuronal lines without the 16p11.2 deletion (*n* = 6 biological replicates; Fig. 2a). We then asked whether, on average, the 200 neuronally expressed genes on 16p were differentially expressed in response to the 16p11.2 deletion (Methods). Genes on 16p had significantly lower expression in the deletion lines (mean log₂(fold-change) = −0.015, *P* = 0.02; mean fold-change *t*-statistic = −0.16, *P* = 0.01; Fig. 2b). The deletion's effect on 16p genes was different from the effect on all other 8,533 neuronally expressed genes in the genome (*P* = 0.02), the expression of which was not on average changed by the deletion (mean fold-change *t*-statistic = −0.01, *P* = 0.44) (Fig. 2c). In contrast, for the 189 genes on 16p with lower baseline neuronal expression (below all-gene median), expression did not significantly change in response to the 16p11.2 deletion (*P* > 0.2 for all comparisons; Supplementary Fig. 16). This analysis suggests that one of the most common autism-associated deletions is associated with transcriptional perturbation of genes in the surrounding region.

Recurrent deletions at 15q13.3 are also observed in autism[13,38–40]. To explore the specificity of our findings at 16p11.2, we explored the consequences of deletion of 15q13.3 in the same isogenic neuronal model (*n* = 11 heterozygous deletion replicates, *n* = 6 controls; Methods). In contrast to 16p11.2, 15q13.3 was not associated with transcriptional perturbation of 100 neuronally expressed genes in the surrounding region (mean log₂(fold-change) = −0.01, *P* = 0.54; mean fold-change *t*-statistic = 0.09, *P* = 0.42) and was not different from the effect on all other 8,087 neuronally expressed genes (*P* = 0.37) (Supplementary

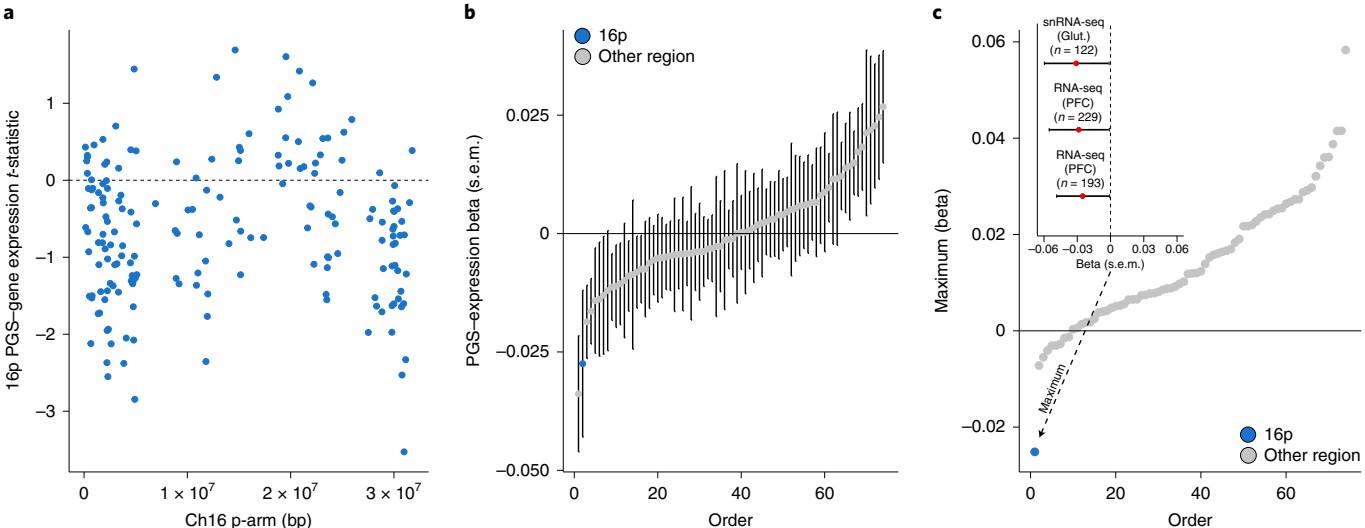

**Fig. 3 | The 16p PGS is associated with decreased average expression of neuronally expressed genes on 16p. a**, Per-gene association to 16p autism PGS in combined analysis of HBTRC and CommonMind resources (*n* = 544 samples). Each point is a gene expressed in glutamatergic neurons (*n* = 183; Methods), the *x* position its midpoint and the *y* position its *t*-statistic from a linear model of normalized expression on 16p autism PGS, controlling for donor schizophrenia diagnosis, European ancestry and single-cell/bulk expression measurement. **b**, Association between average regional gene expression and regional PGS across 16p (blue point) and 73 other 33-Mb control regions (gray points). Point estimates and s.e.m. are from regression of average regional gene expression on regional autism PGS, controlling for donor schizophrenia diagnosis, European ancestry and single-cell/bulk expression measurement (*n* = 544 samples). **c**, Consistency of association between regional autism PGS and average gene expression across the genome. Inset: association between mean 16p gene expression and autism 16p PGS in each of the three contributing cohorts: HBTRC snRNA-seq (glutamatergic neurons (Glut.)), *n* = 122 European ancestry samples; CommonMind bulk RNA-seq (cortical tissue (PFC)), *n* = 229 European ancestry samples; and CommonMind bulk RNA-seq (cortical tissue), *n* = 193 African ancestry samples. Point estimates and s.e.m. are from regression of mean 16p gene expression on 16p autism PGS, controlling for donor schizophrenia diagnosis. The 16p region exhibits the most consistent negative association between PGS and gene expression across the three cohorts compared with other 33-Mb regions. The *y* axis is the most positive regression coefficient in the model described in **c** (inset). Analysis is repeated for the other 33-Mb control regions (gray points).

Fig. 17). These results suggest that the transcriptional observations at 16p11.2 are not an artifact of the CRISPR-mediated deletion because the 15q13.3 and 16p11.2 models share experimental design. These results also suggest that the regional transcriptional effects observed at 16p11.2 are not shared across all autism-associated CNVs.

**The 16p autism PGS associates with decreased 16p gene expression**

Our analysis of 16p11.2 deletion lines suggests that this genetic event causes transcriptional perturbation across 16p. Given that our S-pTDT analysis identified excess polygenic influence on autism across 16p, we tested the hypothesis that this common variant factor would also associate with decreased mean expression across 16p.

We analyzed paired genotype and expression data from three sources. First, we drew on data from ongoing single-nucleus RNA-sequencing (snRNA-seq) analysis of prefrontal cortex (Brodmann area 46) from 122 European ancestry donors from the Harvard Brain Tissue Resource Center/National Institutes of Health (NIH) NeuroBioBank (HBTRC) (Supplementary Fig. 18); we performed our analyses in glutamatergic neurons because they were the most abundant cell type and the most similar to the induced neurons from the in vitro deletion analysis[41]. Next, we analyzed paired genotype and bulk cortical RNA-seq from the CommonMind Consortium, split into two ancestry-specific subgroups (*n* = 193 individuals of African ancestry, *n* = 229 individuals of European ancestry) (Supplementary Fig. 19)[42]. Both the HBTRC and the CommonMind cohorts included donors with and without schizophrenia, and we controlled for this diagnostic status in our analyses. Within each cohort, we constructed regional PGSs for autism within the 33-Mb partitions described above and regressed average regional gene expression on the regional PGS (Methods). To increase power, and to be consistent across datasets, we restricted each of the three analyses to half the genes with the highest expression in glutamatergic neurons in the HBTRC data (*n* = 8,878 genes; Methods).

Increased autism PGS within 16p was associated with decreased expression of genes throughout the 16p region (Fig. 3a; *n* = 183 genes, combined cohort permutation *P* = 0.03; Methods). Relative to the 73 other control regions, 16p had the second most negative association between regional PGS and mean gene expression (Fig. 3b). In addition, 16p exhibited by far the most consistently negative association between PGS and gene expression across the three cohorts (Fig. 3c). We performed two additional sensitivity analyses: first, we found a weaker association in the half of genes with lower expression in glutamatergic neurons (Supplementary Fig. 20) and, second, we showed the association to be robust to an alternative approach to controlling for sample ancestry (Supplementary Fig. 21). In summary, we observed across independent cohorts that increased 16p autism PGS is associated with an average decrease in gene expression within the partition.

**Convergence of gene expression and chromatin contact at 16p**

Given that both the 16p11.2 deletion and 16p autism PGS are associated with decreased average gene expression in 16p, we asked whether these effects converged at the level of individual genes. We found a positive association between the per-gene expression effects of the 16p11.2 deletion and the 16p autism PGS across 168 glutamatergically expressed genes shared across both datasets (*r* = 0.18, *P* = 0.02; Fig. 4a and Supplementary Fig. 22). This observation suggests that the common variant 16p autism PGS and the rare variant 16p11.2 deletion share downstream functional impact on gene expression. We also note that genes with expression decreased in response to both the 16p autism PGS and the 16p11.2 deletion are enriched at the end of 16p (Ch16: 0–5.2 Mb, 'telomeric region'; χ² *P* = 0.003 for negative *t*-statistic in both cohorts and telomeric region location; Methods). This telomeric region of

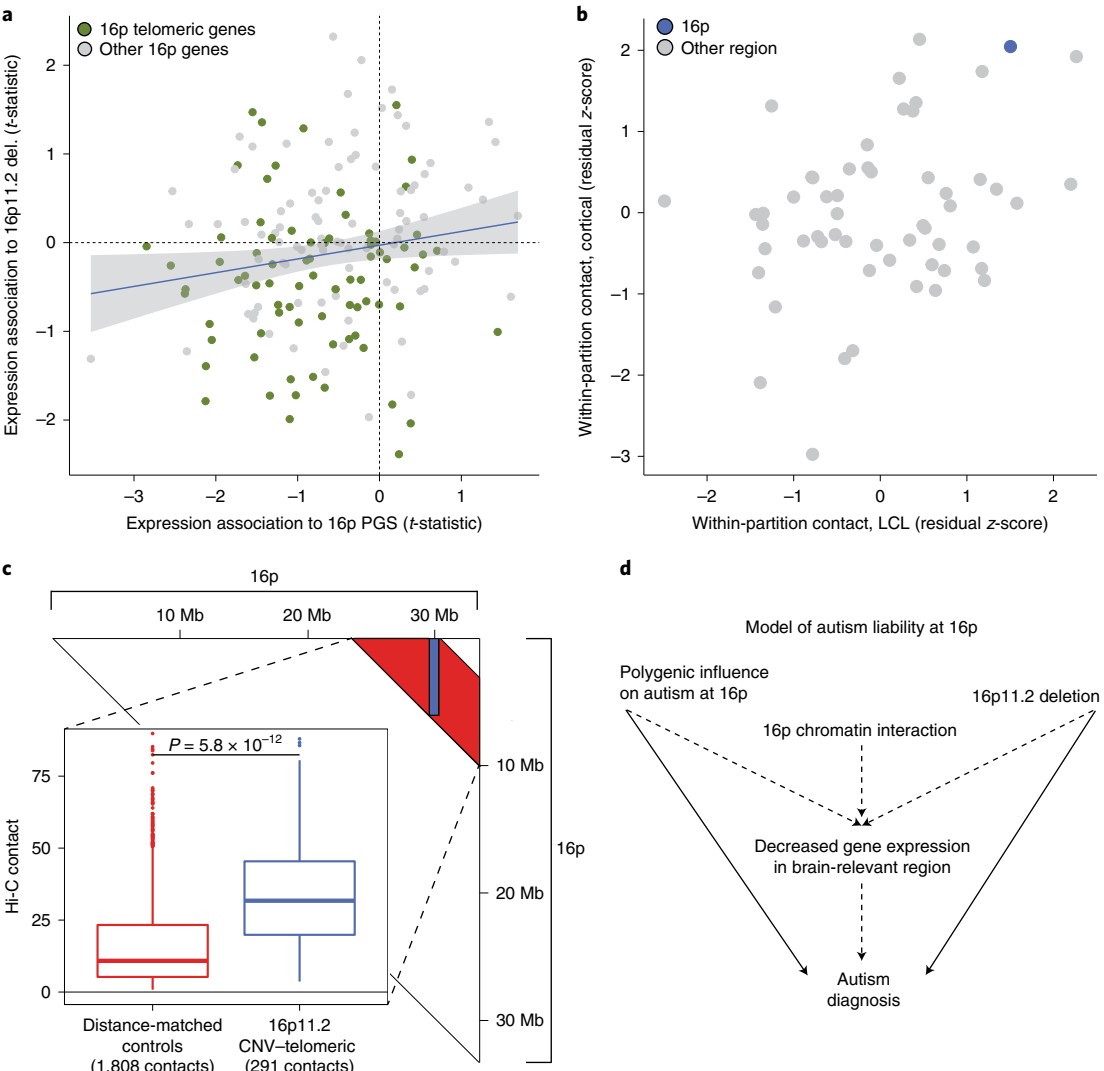

**Fig. 4 | Integrative model of genetic influences on autism at 16p. a**, On the *x* axis, the association *t*-statistics from the all sample meta-analyses of 16p PGS and 16p gene expression; on the *y* axis, the association *t*-statistics from the 16p11.2 in vitro deletion analysis. The shaded region is the 95% CI. Genes are colored by their location on 16p; telomeric is defined as a gene midpoint <5.2 Mb. A single outlying point has been truncated from the plot for visualization; the untruncated plot is shown in Supplementary Fig. 22. **b**, Hi-C analysis revealing elevated within-region chromatin interaction at 16p. Each point represents a 33-Mb partition, with 16p colored blue. Both axes are in units of residual *z*-score, where the residual is from a linear model regressing out segmental duplication content and gene count from the mean within-region Hi-C contact value (Methods). The *x* axis is a dataset of LCLs, whereas the *y* axis is a dataset of midgestational cortical plate. **c**, Hi-C analysis revealing elevated contact between the 16p11.2 locus and the 5.2-Mb gene-dense telomeric region of 16p in

midgestational cortical plate. The triangle depicts the 16p contact matrix: the blue shaded region denotes contacts between the 16p11.2 locus (29.5–30.2 Mb) and the 0- to 5.2-Mb telomeric region (*n* = 291 100-kb × 100-kb contacts), whereas the red shaded region is the distance between matched controls (*n* = 1,808 100-kb × 100-kb contacts). The inset shows the distribution of contact values for 16p11.2–telomeric versus control contacts. The *P* value is from a two-sided, two-sample Student's *t*-test. The lower whisker, lower hinge, center, upper hinge and upper whisker correspond to (lower hinge − 1.5× interquartile range (IQR)) and the 25th percentile, median, 75th percentile, and (upper hinge + 1.5× IQR), respectively. **d**, A model of genetic influences on autism at 16p. Two independent genetic influences on autism—the 16p11.2 deletion and polygenic variation at 16p—are located in a region of elevated 16p chromatin interaction and enriched in brain-specific expression and are associated with coordinated decreased gene expression at 16p.

chromosome 16 is very gene dense—with 182 genes, it is the second most gene dense of 526 5.2-Mb regions in the genome. As with 16p more broadly, it is enriched in genes specifically expressed in adult cortex (*n* = 33, 83% more than expected by chance, *P* = 0.0002).

The correlation in transcriptional effects associated with the 16p PGS and 16p11.2 deletion motivated us to explore genomic structural factors that could help to explain these coordinated effects across a large segment of the genome. We hypothesized that the 16p region may have increased within-region chromatin contact, which could explain the apparent nonindependence of genetic and expression variation on megabase scale. To examine chromatin contact within 16p, we used two published Hi-C datasets: a dataset of lymphoblastoid cell lines

(LCLs)[43] and a dataset from the primarily neuronal midgestational cortical plate[44]. The *i,j*th entry of a Hi-C contact matrix estimates the degree of physical interaction between the *i*th and *j*th regions of the genome. We estimated contact within 33-Mb partitions as the mean of the off-diagonal values of the contact matrix. As segmental duplication content and gene density of the partition are associated with mean Hi-C estimates (Supplementary Fig. 23), we regressed them out to yield a per-partition *z*-score, which we interpreted as the Hi-C regional contact corrected for these genomic features.

The 16p partition exhibits high levels of within-region contact in both cohorts: 4/74 highest partition in LCL (*z*-score: 1.50) and 2/74 highest in the cortical plate dataset (*z*-score: 2.05) (Fig. 4b). We hypothesize

that this diffusely elevated within-region contact at 16p could facilitate the influence of regional polygenic effects on gene expression across 16p, via complex distal regulatory interactions.

Our analysis of the in vitro 16p11.2 deletion neurons (Fig. 2b) revealed decreased gene expression at the gene-dense telomeric region of chromosome 16. We hypothesized that this is because the 16p11.2 locus has increased physical interaction with this telomeric region. Consistent with this hypothesis, in midgestational cortical plate Hi-C data, the 16p11.2–telomeric contacts ($n = 291$ 100-kb × 100-kb contacts) are 2.9× more frequent than contacts between distance-matched control regions on 16p ($n = 1,808$ 100-kb × 100-kb contacts, $P = 5.8 \times 10^{-12}$; Fig. 4c and Methods). In conclusion, these results suggest that the three-dimensional conformation of 16p may mediate convergent autism-related genetic effects on gene expression via regulatory interactions across megabases of separation.

## Discussion

Our observations motivate us to hypothesize the following model (Fig. 4d): genetic influence on autism emerges from the well-established 16p11.2 deletion and from common polygenic variation that is distributed across the region. Both of these influences are associated with an average decrease in cortically expressed genes across 16p and their expression effects are correlated at the gene level. We hypothesize that these transcriptional changes increase the likelihood of an autism diagnosis subsequent to the unusually large number of genes specifically expressed in the cortex at 16p. We also hypothesize that the region's elevated internal chromatin contact may facilitate the transcriptional convergence of these two distinct influences. This hypothesis is consistent with work demonstrating that both single nucleotide[28] and structural[25-27] variation can cause transcriptional and chromatin perturbation. The distributed effect is also consistent with the results of a recent large-scale, exome-sequencing study of autism, which found that no single gene within the 16p11.2 locus was strongly associated with autism[8]. Our model adds to a literature of multi-gene[22] and genetic network effects[20,23,24,45] associated with the 16p11.2 CNV and integrates common variation and chromatin architecture with 16p11.2 and the broader 33-Mb 16p region.

Our analysis of large regions of genome is noncanonical in complex trait genetics, contrasting with a common approach focused on mapping disease-associated variants to the genes through which they act[4-6]. Existing approaches such as transcriptome-wide association studies aggregate individually modest genetic effects on expression to associate genes with phenotype[46]. In the present study, we aggregate both genetic effects on expression and effects across many genes in a region, increasing power to observe modest effects. Regional analysis also allows new perspectives into gene function, including the observation of a region enriched in genes specifically expressed in the brain or enriched in chromatin contact. Our results suggest that chromatin landscapes can facilitate convergent genetic and transcriptional effects within large regions of the genome. This insight supports the viability of a new approach for extracting biological insight from genetic association data across large genomic regions.

Our observations raise many questions for future study. Why are the genetic and transcriptional associations at 16p related to autism? On the one hand, we found that the region harbors an unusual concentration of genes specifically expressed in the brain, but, on the other, not an unusual number of genes implicated in autism from exome association studies. We did not find a 16p signal in ADHD trios using the S-pTDT analysis, arguing against viewing 16p as equally relevant across neurodevelopmental traits. It is also possible that the genic relevance of the region will become apparent only through analysis of the biological networks into which 16p proteins interact and integrate; growing resources of protein–protein interaction data will facilitate this line of inquiry[47]. The mean expression effects are modest, especially compared with the decrease in gene expression associated with heterozygous gene deletion such as that seen with the 16p11.2 CNV. Future studies will probe the biological consequence of modest expression changes spread across many genes. This analysis also raises the question of whether there are other regions of the genome in which common and rare variation converge in a similar fashion with relevance for either autism or other traits. In conclusion, our analysis presents a new statistical approach for partitioned polygenic association and uncovers surprising functional convergence of common and rare variant influences on autism at 16p.

## Online content

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

[1]Stanley Center for Psychiatric Research, Broad Institute of MIT and Harvard, Cambridge, MA, USA. [2]Department of Biomedical Informatics, Harvard Medical School, Boston, MA, USA. [3]Department of Genetics, Harvard Medical School, Boston, MA, USA. [4]Program in Medical and Population Genetics, Broad Institute of MIT and Harvard, Cambridge, MA, USA. [5]Department of Neurology, Massachusetts General Hospital and Harvard Medical School, Boston, MA, USA. [6]Center for Genomic Medicine, Massachusetts General Hospital, Boston, MA, USA. [7]Center for Genomics and Personalized Medicine, Aarhus University, Aarhus, Denmark. [8]Department of Biomedicine (Human Genetics) and iSEQ Center, Aarhus University, Aarhus, Denmark. [9]Bioinformatics Research Centre, Aarhus University, Aarhus, Denmark. [10]The Lundbeck Foundation Initiative for Integrative Psychiatric Research, iPSYCH, Aarhus, Denmark. [11]Analytic and Translational Genetics Unit, Massachusetts General Hospital, Boston, MA, USA. [12]Center for Neonatal Screening, Department for Congenital Disorders, Statens Serum Institut, Copenhagen, Denmark. [13]Department of Psychiatry, Harvard Medical School, Boston, MA, USA. [14]McLean Hospital, Belmont, MA, USA. [15]Department of Psychiatry, Massachusetts General Hospital and Harvard Medical School, Boston, MA, USA. [16]Department of Psychiatry, University of California San Diego, La Jolla, CA, USA. *Lists of authors and their affiliations appear at the end of this paper. ✉e-mail: dweiner@broadinstitute.org; erob@broadinstitute.org

## iPSYCH Consortium

**Preben B. Mortensen**[10,17,18], **Thomas Werge**[10,19,20], **Ditte Demontis**[7,8,10], **Ole Mors**[10,21], **Merete Nordentoft**[10,22], **Thomas D. Als**[7,8,10], **Marie Baekvad-Hansen**[10,12] and **Anders Rosengren**[10,19]

[17]National Centre for Register-Based Research, Aarhus University, Aarhus, Denmark. [18]Centre for Integrated Register-based Research, Aarhus University, Aarhus, Denmark. [19]Institute of Biological Psychiatry, MHC Sct. Hans, Mental Health Services Copenhagen, Roskilde, Denmark. [20]Department of Clinical Medicine, University of Copenhagen, Copenhagen, Denmark. [21]Psychosis Research Unit, Aarhus University Hospital, Aarhus, Denmark. [22]Mental Health Services in the Capital Region of Denmark, University of Copenhagen, Copenhagen, Denmark.

## ASD Working Group of the Psychiatric Genomics Consortium

**Alexandra Havdahl**[23], **Anders Rosengren**[10,19], **Anders D. Børglum**[7,8,10], **Anne Hedemand**[8], **Aarno Palotie**[24], **Aravinda Chakravarti**[25], **Elise B. Robinson**[1,6,15], **Dan Arking**[26], **Arvis Sulovari**[27], **Anna Starnawska**[8], **Bhooma Thiruvahindrapuram**[28], **Christiaan de Leeuw**[29], **Caitlin Carey**[1,6,11], **Christine Ladd-Acosta**[30], **Celia van der Merwe**[11], **Bernie Devlin**[31], **Edwin H. Cook**[32], **Evan Eichler**[33], **Elisabeth Corfield**[34], **Gwen Dieleman**[35], **Gerard Schellenberg**[36], **Jakob Grove**[7,8,9,10], **Hakon Hakonarson**[37], **Hilary Coon**[38], **Isabel Dziobek**[39], **Jacob Vorstman**[40], **Jessica Girault**[41], **Jack M. Fu**[1,4,5,6], **James S. Sutcliffe**[42], **Jinjie Duan**[8], **John Nurnberger**[43], **Joachim Hallmayer**[44], **Joseph Buxbaum**[45], **Joseph Piven**[41], **Lauren Weiss**[46], **Lea Davis**[47], **Magdalena Janecka**[48], **Manuel Mattheisen**[10], **Matthew W. State**[46], **Michael Gill**[49], **Mark Daly**[11], **Mohammed Uddin**[50], **Ole Andreassen**[51], **Peter Szatmari**[40], **Phil Hyoun Lee**[1], **Richard Anney**[52], **Stephan Ripke**[53], **Kyle Satterstrom**[1], **Susan Santangelo**[54], **Susan Kuo**[1], **Ludger Tebartz van Elst**[55], **Thomas Rolland**[56], **Thomas Bougeron**[56], **Tinca Polderman**[57], **Tychele Turner**[58], **Jack Underwood**[59], **Veera Manikandan**[60], **Vamsee Pillalamarri**[26] and **Varun Warrier**[61]

[23]Department of Psychology, University of Oslo, Oslo, Norway. [24]Institute for Molecular Medicine, University of Helsinki, Helsinki, Finland. [25]Center for Human Genetics and Genomics, New York University, New York, NY, USA. [26]McKusick–Nathans Institute of Genetic Medicine, Johns Hopkins University, Baltimore, MD, USA. [27]Cajal Neuroscience, Seattle, WA, USA. [28]The Centre for Applied Genomics, The Hospital for Sick Children, Toronto, Ontario, Canada. [29]Department of Complex Trait Genetics, VU University, Amsterdam, the Netherlands. [30]Bloomberg School of Public Health, Johns Hopkins University, Baltimore, MD, USA. [31]Department of Psychiatry, University of Pittsburgh, Pittsburgh, PA, USA. [32]Department of Psychiatry, University of Illinois at Chicago, Chicago, IL, USA. [33]Department of Genome Sciences, University of Washington, Seattle, WA, USA. [34]Department of Mental Disorders, Norwegian Institute of Public Health, Oslo, Norway. [35]Department of Child and Adolescent Psychiatry, Sophia Childrens Hospital, Rotterdam, the Netherlands. [36]Department of Pathology and Laboratory Medicine, University of Pennsylvania School of Medicine, Philadelphia, PA, USA. [37]The Center for Applied Genomics and Division of Human Genetics, University of Pennsylvania School of Medicine, Philadelphia, PA, USA. [38]Department of Psychiatry, University of Utah, Salt Lake City, UT, USA. [39]Department of Psychology, Humboldt Universitat Berlin, Berlin, Germany. [40]Department of Psychiatry, University of Toronto, Toronto, Ontario, Canada. [41]Department of Psychiatry, UNC School of Medicine, Chapel Hill, NC, USA. [42]Center for Human Genetics Research, Vanderbilt University, Nashville, TN, USA. [43]Department of Psychiatry, Indiana University School of Medicine, Indianapolis, IN, USA. [44]Department of Psychiatry, Stanford University, Stanford, CA, USA. [45]Department of Neuroscience, Icahn School of Medicine at Mount Sinai, New York, NY, USA. [46]Department of Psychiatry, University of California, San Francisco, San Francisco, CA, USA. [47]Vanderbilt Genetics Institute, Vanderbilt University Medical Center, Nashville, TN, USA. [48]Department of Psychiatry, Icahn School of Medicine at Mount Sinai, New York, NY, USA. [49]Department of Psychiatry, Trinity College Dublin, Dublin, Ireland. [50]College of Medicine, Mohammed Bin Rashid University of Medicine and Health Sciences, Dubai, UAE. [51]Institute of Clinical Medicine, University of Oslo, Oslo, Norway. [52]Division of Psychological Medicine and Clinical Neurosciences, Cardiff University, Cardiff, UK. [53]Department of Psychiatry and Psychotherapy, Charite, Berlin, Germany. [54]Center for Psychiatric Research, Maine Medical Center Research Institute, Portland, ME, USA. [55]Department of Psychiatry and Psychotherapy, University of Freiburg, Freiburg, Germany. [56]Department of Neuroscience, Institut Pasteur, Paris, France. [57]Department of Child and Adolescent Psychiatry, University of Amsterdam, Amsterdam, the Netherlands. [58]Department of Genetics, University of Washington School of Medicine in St. Louis, St. Louis, MO, USA. [59]Cardiff University, Cardiff, UK. [60]Regeneron, White Plains, NY, USA. [61]Department of Psychiatry, University of Cambridge, Cambridge, UK.

## ADHD Working Group of the Psychiatric Genomics Consortium

**Alexandra Philipsen**[62], **Andreas Reif**[63], **Anke Hinney**[64], **Bru Cormand**[65,66,67,68], **Claiton H. D. Bau**[69,70], **Diego Luiz Rovaris**[71], **Ditte Demontis**[7,8,10], **Edmund Sonuga-Barke**[72], **Elizabeth Corfield**[34,73], **Eugenio Horacio Grevet**[74,75], **Giovanni Salum**[74,76], **Henrik Larsson**[77,78], **Jan Buitelaar**[79], **Jan Haavik**[80,81], **James McGough**[82], **Jonna Kuntsi**[72], **Josephine Elia**[83,84], **Klaus-Peter Lesch**[85,86,87], **Marieke Klein**[16], **Mark Bellgrove**[88], **Martin Tesli**[34], **Patrick W. L. Leung**[89], **Pedro M. Pan**[90], **Soren Dalsgaard**[91,92,93], **Sandra Loo**[82], **Sarah Medland**[94], **Stephen V. Faraone**[95], **Ted Reichborn-Kjennerud**[51,96], **Tobias Banaschewski**[97] and **Ziarih Hawi**[88]

[62]Department of Psychiatry and Psychotherapy, University Hospital Bonn, Bonn, Germany. [63]Department for Psychiatry, Psychosomatic Medicine and Psychotherapy, University Hospital Frankfurt, Frankfurt, Germany. [64]Department of Child and Adolescent Psychiatry, University Hospital Essen, Essen, Germany. [65]Department of Genetics, Microbiology and Statistics, University of Barcelona, Barcelona, Spain. [66]Centro de Investigación Biomédica en Red de Enfermedades Raras, Salamanca, Spain. [67]Institut de Biomedicina de la Universitat de Barcelona, Barcelona, Spain. [68]Institut de Recerca Sant Joan de Déu, Barcelona, Spain. [69]Department of Genetics, Universidade Federal do Rio Grande do Sul, Porto Alegre, Brazil. [70]ADHD and Developmental Psychiatry Programs, Hospital de Clínicas de Porto Alegre, Porto Alegre, Brazil. [71]Instituto de Ciencias Biomedicas, Universidade de Sao Paulo, São Paulo, Brazil. [72]Institute of Psychiatry, Psychology and Neuroscience, King's College London, London, UK. [73]Nic Waals Institute, Lovisenberg Diaconal Hospital, Oslo, Norway. [74]Department of Psychiatry, Universidade Federal do Rio Grande do Sul, Porto Alegre, Brazil. [75]Adult ADHD Outpatient Program, Hospital de Clínicas de Porto Alegre, Porto Alegre, Brazil. [76]Section on Negative Affect and Social Processes, Hospital de Clínicas de Porto Alegre, Porto

Alegre, Brazil. [77]School of Medical Sciences, Örebro University, Örebro, Sweden. [78]Department of Medical Epidemiology and Biostatistics, Karolinska Institutet, Stockholm, Sweden. [79]Department of Cognitive Neuroscience, Donders Institute for Brain, Cognition and Behavior, Nijmegen, the Netherlands. [80]Department of Biomedicine, University of Bergen, Bergen, Norway. [81]Division of Psychiatry, Haukeland University Hospital, Bergen, Norway. [82]Semel Institute for Neuroscience and Human Behavior, UCLA David Geffen School of Medicine, Los Angeles, CA, USA. [83]Department of Pediatrics, Nemours Children's Hospital, Wilmington, DE, USA. [84]Sidney Kimmel Medical College, Thomas Jefferson University, Philadelphia, PA, USA. [85]Division of Molecular Psychiatry, University of Würzburg, Würzburg, Germany. [86]Institute of Molecular Medicine, I.M. Sechenov First Moscow State Medical University, Moscow, Russia. [87]Department of Neuropsychology and Psychiatry, Maastricht University, Maastricht, the Netherlands. [88]Turner Institute for Brain and Mental Health, Monash University, Melbourne, Australia. [89]Department of Psychology, The Chinese University of Hong Kong, Hong Kong, China. [90]Department of Psychiatry, Federal University of São Paulo, São Paulo, Brazil. [91]Department of Child and Adolescent Psychiatry, Mental Health Services of the Capital Region, Copenhagen, Denmark. [92]Institute of Clinical Medicine, University of Copenhagen, Copenhagen, Denmark. [93]National Centre for Register-based Research, Aarhus University, Aarhus, Denmark. [94]Psychiatric Genetics, QIMR Berghofer Medical Research Institute, Brisbane, Australia. [95]Departments of Psychiatry and Neuroscience and Physiology, SUNY Upstate Medical University, Syracuse, NY, USA. [96]Norwegian Institute of Public Health, Oslo, Norway. [97]Department of Child and Adolescent Psychiatry, Heidelberg University, Heidelberg, Germany.

## Methods

We confirm that the present study was reviewed and approved by Massachusetts General Brigham institutional review board (IRB). The study name is Molecular Study of Cognitive and Behavioral Variation (IRB: 2015P002376). The principal investigator was E.B.R. The iPSYCH study was approved by the Danish Data Protection Agency and the Scientific Ethics Committee in Denmark.

### Generation of PGSs

For autism PGS analysis in autism trios, we used a GWAS from the iPSYCH collection in Denmark because there is no sample overlap with the autism trio samples (19,870 individuals with autism, 39,078 controls; Supplementary Table 2). For all other autism PGS analysis (for example, PGS–expression analyses), we used a meta-analysis of the same iPSYCH autism samples, plus autism samples from the PGC (combined: 26,067 individuals with autism, 46,455 controls). For analysis of ADHD, we used a nonoverlapping iPSYCH-only ADHD GWAS (25,895 individuals with ADHD, 37,148 controls).

To generate PGS weights, we first applied LDpred v.1.0.11 on the marginal effect sizes from GWASs[48]. We used LDpred under the infinitesimal genetic architecture model with LD reference from Hapmap 3 SNPs ($n$ = 503 European ancestry samples). All PGSs were calculated using the --score function in PLINK 1.9 (ref. [49]). As LDpred estimates posterior causal effect sizes from GWAS marginal effect sizes, we include all SNPs in PGS analysis, including when constructing stratified PGSs for S-pTDT.

### Autism family cohorts

The collection, imputation and quality control of the SSC and Simons Foundation Powering Autism Research (SFARI) have been described previously (Supplementary Table 3)[12,50]. The autism trios from the PGC Autism group (PGC) are as described previously[12], with the modification of the inclusion of probands from multiplex families. We defined a European ancestry subset of PGC for analysis by generating principal components of ancestry using PLINK 1.9 and by visual inspection relative to HapMap reference populations (Supplementary Fig. 1). We defined a family as European ancestry if both parents and proband were of European ancestry by principal component analysis (PCA) (4,335 of 5,283 trios, 82%).

### Genome-wide pTDT

We performed genome-wide pTDT to assess power for S-pTDT analyses in SSC, SPARK and PGC. We estimated polygenic transmission as described previously[12], with the exception of an adapted approach for the case/pseudocontrol genotypes in PGC (Supplementary Note). The results for each of the three cohorts are displayed in Supplementary Fig. 2.

### S-pTDT

S-pTDT is identical to pTDT, except that, instead of testing for transmission of a PGS constructed from all SNPs, it tests for transmission of a PGS constructed from a subset of SNPs:

$$S-pTDT = \left(PGS_{S,C} - PGS_{S,MP}\right)/s.d.\left(PGS_{S,MP}\right)$$

where $PGS_{S,C}$ is the stratified PGS of child C and $PGS_{S,MP}$ is the mid-parent-stratified PGS (average of the two parents). S-pTDT is a one-sample two-sided Student's $t$-test for whether the S-pTDT distribution has a mean different from 0. S-pTDT estimates of a given PGS for a given cohort is equal to the average S-pTDT value for all families in the cohort.

We created stratified PGSs by dividing SNPs into sets of equal sizes. We varied this partitioning in two ways to complete a comprehensive survey of regional transmission: first, we divided the SNPs into partitions of varying sizes (2,000, 3,000, 4,000, 5,000 and 6,000 SNPs) and,

second, we started the partitioning from either the beginning or the end of the chromosome. For example, for creating genomic blocks of 2,000 SNPs, we identified the first PGS SNP on chromosome 1 (the SNP closest to the first basepair), counted 2,000 PGS SNPs and defined this as the first partition. Then, we counted the next 2,000 PGS SNPs on chromosome 1, defined this as the next partition, until there were fewer than 2,000 SNPs remaining on chromosome 1. Next, we repeated the same procedure on chromosome 2 and for all remaining chromosomes. By varying the number of SNPs in the partition and whether SNP counting began at the start or the end of the chromosome, we produced 2,006 (often overlapping) partitions (1,003 from the start of chromosomes, 1,003 from the end of chromosomes). Before partitioning, we subsetted the PGS SNPs to those present in all three autism trios cohorts (SSC, SPARK and PGC) to avoid bias from SNP missingness across partitions. We then estimated stratified PGSs for each partition in each of the three cohorts using linear scoring (--score) in PLINK 1.9 and performed S-pTDT on each partition as described above.

Partition length and SNP count were predictive of over-transmission (Supplementary Fig. 4). We regressed out expected over-transmission using a linear model—S-pTDT ≈ (number of PGS SNPs) + (length of partition in basepairs)—and normalized the model residuals by the s.d. of the model residual distribution. This procedure yields for each partition a residual $z$-score, which estimates the number of s.d.s by which the partition is over- or under-transmitted relative to expectation. If partitions included a gap between adjacent SNPs >1 Mb, we adjusted the contribution of that gap down to 1 Mb, which accounts for decay in LD but avoids inappropriately correcting the S-pTDT signal in the over-transmission model noted above. For analysis of 33-Mb partitions, the S-pTDT $z$-score regressed out only SNP number, because the basepair length of all partitions was the same.

### Supplementary analyses for 16p S-pTDT association

First, we confirmed that autism-associated loci through GWAS were enriched in the S-pTDT distribution. We defined an autism-associated locus as the five loci from the most recently published autism GWAS reaching genome-wide significance from analysis of autism alone (index SNPs: rs910805, rs10099100, rs201910565, rs71190156 and rs111931861)[11].

Next, we analyzed the distribution of autism-associated CNVs in the S-pTDT distribution. We identified autism-associated CNVs from the set on SFARI Gene (https://gene.sfari.org/database/cnv) and then identified the S-pTDT partitions with at least one of these CNVs within the boundary (16p11.2, 16p13.11, 16p13.3, 16p12.2, 2p16.3, 15q13.3, 7q11.23, 17q12, 3q29, 1q21.1, 17p11.2, 8p23.1, 17q11.2, 2q11.2, 22q11.2, 22q13.3 and 5q35; Supplementary Fig. 10). We also estimated the association between segmental duplication content and S-pTDT for the 33-Mb partitions: we annotated each partition for segmental duplication rate by calculating the fraction of nucleotides in each partition that overlapped at least one segmental duplication per the University of California, Santa Cruz (UCSC) Genome Browser[51]. Coverage calculations were performed using BEDTools v.2.30.0 (Supplementary Fig. 11)[52].

To rule out the contribution of 16p CNV carriers driving the S-pTDT signal, we repeated S-pTDT analysis in SSC + SPARK after removing trios where the proband carried an inherited or de novo neurodevelopmental disorder-associated CNV at 16p (we could not perform this analysis in PGC because we did not have exome sequencing for this cohort). We adopted a literature-based definition of neurodevelopmental disorder-associated CNVs from a recent autism sequencing study[8]. Of the 5,048 trios in the SSC + SPARK analysis, we removed 51 (1.0%) with a qualifying CNV and repeated the S-pTDT analysis (Supplementary Fig. 9).

We tested the hypothesis that S-pTDT relates to density of accessible chromatin in the developing human brain as tagged by H3K27ac histone marks. We analyzed a published resource of chromatin

immunoprecipitation (ChIP)–sequencing profiling of two biological replicates of 7-week post-conception human cortex[33]. Within each replicate, we summed the count of H3K27ac peaks within each of our defined 33-Mb partitions, scaled the counts to mean = 0 and s.d. = 1, and then averaged the $z$-scores between the two replicates.

To evaluate the specificity of the S-pTDT finding on 16p, we performed an analogous analysis in ADHD. We used 1,634 European ancestry ADHD trios from the PGC and an external ADHD GWAS from the iPSYCH consortium with 25,895 individuals with ADHD and 37,148 controls[53]. We partitioned the genome into blocks of 2,000, 3,000, 4,000, 5,000 and 6,000 SNPs as described above, starting from the beginning of chromosomes, and estimated the S-pTDT for each partition. We then estimated a residual $z$-score, regressing out the number of SNPs and partition size as in the autism analysis (Supplementary Fig. 12).

We next evaluated whether the polygenic signal at 16p could be explained by a specific locus within the region. To perform this analysis, we partitioned 16p into 25 LD-independent blocks, each of approximately 1.5 Mb in size, as previously defined[31]. We then estimated S-pTDT using the iPSYCH-only autism summary statistics in SSC and SPARK for each of these 25 blocks (Supplementary Fig. 8). To evaluate the contribution of individual loci, we estimated the decay in 16p S-pTDT signal as a function of removing the most over-transmitted remaining S-pTDT blocks. Specifically, we (1) estimated per block transmission, (2) ranked the blocks from most to least over-transmission, (3) estimated over-transmission using SNPs from all blocks, (4) estimated over-transmission using SNPs from all blocks minus SNPs from the most associated remaining block and (5) repeated step 4 until only a single block remained (Fig. 1e). For example, the first block ('number of 16p partitions removed from PGS = 0') includes all the 7,658 SNPs in the 16p PGS. The next block ('number of 16p partitions removed from PGS = 1') subtracts 287 SNPs from the most associated block in 16p, leaving this new block with 7,371 SNPs.

We next evaluated the regional polygenic signal of 16p relative to equally sized comparison partitions across the genome. As 16p spans approximately 33 Mb of the genome, we constructed control partitions of 33 Mb by starting at the beginning of chromosomes and defining adjacent 33-Mb blocks (Supplementary Table 4). We defined the start coordinate of a chromosome by the minimum of (1) the first SNP in 1000 Genomes Phase 3 EUR and (2) the start position of first gene in gnomAD[54,55]. Similarly, we defined the end coordinate of a chromosome by the maximum of (1) the last SNP in 1000 Genomes Phase 3 EUR and (2) and end position of the last gene in gnomAD. This approach yielded 74 partitions, including 16p. We performed S-pTDT using these boundaries by constructing stratified PGSs from all SNPs within a given partition.

### Gene density

We first compiled a consensus gene list for gene-density analyses. We defined this consensus list as the intersection of (1) autosomal genes with unique gene names and nonmissing pLI constraint estimates from gnomAD and (2) genes with estimated specific expression in Genotype-Tissue Expression (GTEx) cortex ('Brain_Cortex')[56]. This consensus list included 17,909 genes. We further annotated this list with the 102 genes implicated in autism via exome sequencing in a recent analysis[32]. We then mapped genes to the above-defined 33-Mb boundaries if their gene body midpoint was located within the boundary. We built linear models predicting specific expression in the cortex (top 10% of specific expression $t$-statistic), from the density of all the genes, and calculated two-sided $P$ values from the residual $z$-scores of the regression.

### GO analysis

We performed GO analysis to evaluate enrichment of genes on 16p in annotated biological pathways (http://geneontology.org). We used the same 17,909 genes from the gene-density analysis as

reference genes. We tested for enrichment of all genes on 16p (midpoint <33,000,000 bp, $n$ = 433 genes) across three classes of annotations: biological process, molecular function and cellular component. The results for Bonferroni's significant enrichments in each of the classes of annotations are reported in Supplementary Table 1.

### Autism differential expression analysis

We analyzed whether genes on 16p are overrepresented in differentially expressed genes between individuals diagnosed with autism and controls. We used a previously published differential expression dataset of human brain RNA-seq ($n$ = 51 individuals with autism, $n$ = 936 controls)[34]. We defined genes as differentially expressed at Bonferroni's significant level correcting for the number of genes overlapping between the dataset and our consensus gene list described above ($n$ = 15,288 genes, $n$ = 83 differentially expressed genes, $n$ = 383 genes on 16p). We performed a $\chi^2$ test for enrichment of differentially expressed genes on 16p.

### CRISPR–Cas9-mediated deletion of ASD-associated loci

Design of the CRISPR-mediated deletion of the 16p11.2 loci and differential expression analysis is described in a published resource[37]. Design of the 15q13.3 CNV deletion lines followed the same protocol, with the deletion defined with boundaries Ch15 30,787,764-32,804,328 (GRCh37). We generated $n$ = 11 heterozygous deletion lines, with an additional $n$ = 6 controls exposed to CRISPR construct but not to guide RNA. When estimating the effect of the deletion on the region, we excluded genes ±100 kb of the deletion window, because the *cis*-regulatory regions of these genes may have been perturbed by the creation of the deletion itself.

### PGS–expression analyses

**HBTRC/NIH NeuroBioBank (snRNA-seq).** We generated paired genotype and single-nucleus expression profiles from the dorsolateral prefrontal cortex (DLPFC) of postmortem brain tissue from the HBTRC/NIH NeuroBioBank. The generation of expression profiles will be described in detail in a forthcoming manuscript from the authors of the present report. In brief, we developed and optimized a workflow for creating and analyzing pools of nuclei sampled from brain tissue (DLPFC, BA46) from 20 different donors per pool. In this workflow, we started by dissecting a defined amount of tissue from each donor, obtaining a similar mass of tissue from each specimen while being careful to represent all cortical layers. The frozen tissue samples were then immediately pooled for simultaneous isolation of their nuclei; all subsequent processing steps, including nuclear isolation, encapsulation in droplets and preparation of snRNA-seq libraries, involve all of the donors together. This 'dropulation' workflow allows us to minimize experimental variability, including any technical effects on messenger RNA ascertainment and any effects of cell-free ambient RNA. Each nucleus in these experiments was then reassigned to its donor of origin using combinations of hundreds of transcribed SNPs; although the individual SNP alleles are shared among many donors, the combinations of many SNPs are unique to each donor in the cohort. Nuclei were assigned to seven major cell classes (astrocytes, endothelial cells, γ-aminobutyric acid (GABA)-ergic neurons, glutamatergic neurons, microglia, oligodendrocytes and polydendrocytes) by global clustering and identification of marker genes expressed in each cluster. Median cell-type proportions were: glutamatergic neurons 47.9%, GABA-ergic neurons 18.8%, astrocytes 13.5%, oligodendrocytes 8.0%, polydendrocytes 5.2%, microglia 1.5% and endothelia 1.0%. All downstream analyses used expression data from glutamatergic neurons. The cell type-specific gene-by-donor expression matrices were processed with VST normalization[57].

We performed a number of pre-association quality control (QC) steps. The majority of genotyped samples were European ancestry (1,707/1,770, 96%) and we identified these samples for downstream

analysis using PCA (Supplementary Fig. 18). Next, we identified any samples as expression outliers with mean expression >3 s.d. from the cohort mean (3/125 samples). This yielded a final EUR subset of 122 samples. We identified genes with expression above median across the 122 samples in the count matrix, with the only processing step of normalization of expression count sum equal for all samples. We used these genes (*n* = 8,878) in all subsequent regional PGS–expression analyses.

**CommonMind (bulk cortical RNA-seq).** We next analyzed paired genotype and bulk DLPFC expression data from donors in the CommonMind consortium. Generation of expression count matrices is described in the CommonMind publication[42]. Within CommonMind, we restricted analysis to donors from the National Institute of Mental Health (NIMH) Human Brain Collection Core (HBCC) and the University of Pittsburgh (PITT) biobanks due to previous analysis demonstrating increased concordance with the snRNA-seq resource described above. We performed variance stabilization on the count matrices separately in HBCC and PITT using the SCTransform package in Seurat with the goal of closely paralleling the approach of the single-nucleus resource (parameters: do.scale = FALSE, do.center = FALSE, return.only.var.genes = FALSE, seed.use = NULL, n_genes = NULL)[58].

The CommonMind collection is ancestrally heterogeneous: the two largest groups are African and European ancestry donors. Accordingly, we used PCA to identify donors of African ancestry (*n* = 193) and European ancestry (*n* = 229) and subsequently analyzed each separately (Supplementary Fig. 19). For consistency with the single-nucleus resource, we restricted analysis to donors diagnosed with schizophrenia or controls.

**Per-cohort association and meta-analysis.** For all samples, we calculated regional autism PGC using the largest autism GWAS (iPSYCH + PGC; see Supplementary Table 2) using Plink 1.9 score with a genotype QC (SNP missingness <1%, minor allele frequency >0.1%, imputation INFO >95%).

We performed two classes of local PGS–gene expression association. The first class is a per-gene association, as in Fig. 3a. The second is average gene association, as in Fig. 3b,c. To be consistent across datasets, we restricted all analyses to half the genes most expressed in glutamatergic neurons in the HBTRC data (*n* = 8,878 genes). The association in Fig. 3a is a per-gene association meta-analyzed across the three cohorts. We combined individual-level expression and genotype PGS across the three cohorts; before concatenating the PGSs and expression matrices used in the individual cohort analyses, we within-cohort scaled per-gene expression and per-partition PGS to mean = 0 and s.d. = 1. Per-gene association followed the linear model: gene expression ≈ regional PGS + schizophrenia diagnostic status + ancestry (binary for yes/no African ancestry) + single cell (binary yes/no). The association *t*-statistic is from the regional PGS covariate. For maximum power to detect mean effects, we assessed significance of the mean PGS–expression association using permutation. Specifically, we calculated the mean(*t*-statistic) in 16p, then shuffled the PGS–donor IDs within each cohort, performed association and calculated the mean(*t*-statistic), repeated 1,000×. The permutation *P* value is the number of times the observed PGS was more negative than the permuted PGS.

For the second class of association, we first averaged the gene expression per partition, then performed the association. For per-cohort association, we used the linear model: mean expression of gene ≈ regional PGS + schizophrenia diagnostic status. For combined analysis, we used the linear model: mean gene expression ≈ regional PGS + schizophrenia diagnostic status + ancestry (binary for yes/no African ancestry) + single cell (binary yes/no). We performed a sensitivity analysis for genetic ancestry using the principal components in Supplementary Fig. 21.

## Hi-C analysis

**LCL resource.** Per-chromosome Hi-C count matrices were downloaded from http://hic.umassmed.edu at 1-Mb resolution for GM06990 LCL[43]. As the count matrices were built in hg18, we converted the 33-Mb partitions from hg19 to hg18 using the National Center for Biotechnology Information's (NCBI's) Genome Remapping Service (www.ncbi.nlm.nih.gov/genome/tools/remap). We matched the boundaries of the 33-Mb partitions with their closest 1-Mb cutoffs in the count matrix. For this analysis, we did not analyze partitions spanning centromeres, yielding 56 partitions for analysis. For each partition, we estimated raw within-partition contact frequency as the mean of the off-diagonal elements of the Hi-C count matrix.

**Midgestational cortical plate resource.** We downloaded 0.1-Mb resolution, Hi-C contact matrices from NCBI's Gene Expression Omnibus (GEO) from a resource of midgestational cortical plate samples from three donors[44]. As above, we mapped the boundaries of the 33-Mb partitions to the 0.1-Mb boundaries of the Hi-C matrix. In contrast to the LCLs, the diagonal elements were zeroed out; thus, the estimated raw within-partition contact frequency was estimated as the mean of all elements of the matrix. We analyzed the same 56 partitions as in the LCL analysis.

As our hypothesis pertained to average contact behavior over large regions of the genome, as opposed to more fine-grained analysis of topologically associated domains or gene–enhancer interactions, we analyzed the largest bin window available within each cohort to increase the signal:noise ratio[59].

**Contact model.** Raw within-partition contact frequency varies with gene density and segmental duplication contact (Supplementary Fig. 23). In the LCLs, this covariance is probably due to an increased number of Hi-C reads mapping to regions with increased segmental duplication content. In cortical lines, there are large chunks of zeroed-out elements of the contact matrix, rates of which correlate strongly with segmental duplication content, probably due to intentional zeroing of elements in regions that are difficult to map because of segmental duplication content. Gene density remains a significant predictor of contact frequency after conditioning on segmental duplication content, motivating us to condition on-gene density as well and to extract normalized residuals from the following model: contact frequency ≈ gene count + segmental duplication content. Our primary analysis in Fig. 4b reports these normalized residuals for each partition.

**Telomeric region analysis of 16p11.2 CNV.** We next analyzed the chromatin contact between the 16p11.2 locus and the distal gene-dense start of chromosome 16. We performed this analysis in the midgestational cortical plate data only because the 1-Mb bin resolution of the LCL resource did not have sufficient resolution. We defined the telomeric region based on the gene-dense segment at the start of chromosome 16, from 0 Mb to the closest 100-kb segment after the endpoint of the final brain-expressed gene in that window in the 16p11.2 deletion dataset (5.2 Mb). We defined the 16p11.2 locus as Ch16: 29.5–30.2 Mb. To define control contact regions, we first calculated the minimum (24.3 Mb) and maximum (30.2 Mb) distances spanned by the contact matrix defined by 0–5.2 Mb (telomeric region) and 29.5–30.2 Mb (16p11.2 locus). We then defined control contacts on 16p as all contacts of distance >24.3 or <30.2 that were not located in the telomeric–CNV contact range described above. These results are robust to inclusion of elements of the contact matrix with '0', which probably reflects segmental duplication-rich regions (telomeric–CNV versus control $P < 1 \times 10^{-10}$ for both approaches).

## Reporting summary
Further information on research design is available in the Nature Research Reporting Summary linked to this article.

## Data availability

Individual-level genotypes are available via request to SFARI (https://www.sfari.org) and from the PGC (https://pgc.unc.edu) and its contributing data holders. GWAS summary statistics from iPSYCH are available by request from members of the consortium. RNA-seq data are available from collection 2304 at the NIMH data archive (https://nda.nih.gov/edit_collection.html?id=2304) and the CommonMind Consortium (https://www.nimhgenetics.org/resources/commonmind). Hi-C data are available via the respective referenced publications. GTEx-specific expression data are available from the Price Lab repository (https://alkesgroup.broadinstitute.org/LDSCORE/LDSC_SEG_ldscores/tstats). Additional gene information is available from the gnomAD browser (https://gnomad.broadinstitute.org/downloads).

## Code availability

Relevant customized scripts for polygenic transmission analysis are available at https://github.com/danjweiner/ptdt_16p. Other analyses were performed using R v.4.1 and Python v.3.7.

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

## Acknowledgements

We thank the following for their generous support of this work: the SFARI (grant no. 704413 to E.B.R. and L.J.O.), the NIMH (grant nos. F30MH129009 to D.J.W., R01MH111813 to E.B.R., R01MH115957 to M.E.T. and 1R01MH124851-01 to A.D.B.), the National Institute of General Medical Sciences (grant nos. T32GM007753 and T32GM144273 to A.N. and D.J.W.), the National Institute of Child Health and Development (grant no. R01HD096326 to M.E.T.), the National Institute of Neurological Disorders and Stroke (grant no. R01NS093200 to M.E.T.), the National Library of Medicine (grant no. T15LM007092 to D.J.W.), the Lundbeck Foundation (grant nos. R102-A9118, R155-2014-1724, and R248-2017-2003 to iPSYCH), the Novo Nordisk Foundation (to the Danish National Biobank) and the universities and university hospitals of Aarhus and Copenhagen. High-performance computer capacity for handling and statistical analysis of iPSYCH data on the GenomeDK HPC facility was provided by the Center for Genomics and Personalized Medicine and the Centre for Integrative Sequencing, iSEQ, Aarhus University, Denmark through a grant to A.D.B. Human tissue was obtained from the NIH NeuroBioBank. We thank all the families who participated in the cohorts included in this analysis, without whom this work would not have been possible. We also thank R. Collins and R. Walters for their assistance with these analyses. Finally, we thank the following members of the ADHD Working Group of the PGC: A. R. Hammerschlag, A. Corsico, A. Havdahl, A. Todorov, A. Charach, A. Ashley-Koch, A. Doyle, A. Hervas, A. Miranda, A. Borglum, A. Scherag, A. Thapar, A. Rommel, A. Starnawska, A. Wheeler, A. Rothenberger, A. Arnatkeviciute, B. Franke, B. Neale, C. Liao, C. Hartman, C. Burton, C. Cornforth, C. Bandeira, C. Bau, C. Sanchez, D. Posthuma, F. Cerrato, F. Mulas, F. Degenhardt, G. C. A. Martins, G. P. Stromstad Knudsen, H. C. Steinhausen, H. Hakonarson, H.-C. Steinhausen, H. Roeyers, H.-W. Kim, I. Gizer, I. Waldman, I. Brikell, J. Crosbie, J. Agnew-Blais, J. Martin, J. Gelernter, J. Hebebrand, J. A. Ramos-Quiroga, J. Biederman, J. Sergeant, J. Gamble, J. Pinsonneault, J. Deckert, K. Langley, L. Yang, L. Kent, L. Rohde, M. Mattheisen, M. J. Arranz Calderun, M. Soler Artigas, M. Ribases, M. Mariano, M. Gill, M. O'Donovan, M. Casas, M. Bayes, N. Martin, N. P. Ole Mors, N. Williams, N. Roth Mota, O. A. Andreassen, P. Sham, P. Sullivan, P. Arnold, P. Lichtenstein, P. Rovira, P. Holmans, P. Asherson, P. B. Mortensen, R. Guerra, R. Walters, R. Anney, R. Ebstein, R. Karlsson Linnér, R. Joober, R. Oades, R. Schachar, S. M. Sengupta, S. Johansson, S. H. Witt, S. Nelson, S. Smalley, S. Scherag, T. Zayats, T. Werge, T. Silk, T. Polderman, T. Banaschewski, T. Altar, V. Manikandan, Y. Zhang, Y. Athanasiadis and Y. Wang. Support for the title page creation and format was provided by AuthorArranger, a tool developed at the National Cancer Institute.

## Author contributions

D.J.W. conducted analyses. D.J.W., E.L., S.E., D.J.C.T., R.Y., J.G., J.M.F., C.E.C., N.B., S.B., D.M.H., J.B.-G. and A.D.B. generated data. D.J.W., E.L., S.E., D.J.C.T. and R.Y. designed experiments and tools. D.J.W., E.L., A.N., E.Z.M., J.S., L.J.O., A.D.B., M.E.T., S.A.M. and E.B.R. aided in interpretation of data. E.B.R. supervised the research. D.J.W. and E.B.R. wrote the manuscript.

## Competing interests

M.E.T. consults for BrigeBio Pharma and receives research funding and/or reagents from Illumina Inc., Levo Therapeutics and Microsoft Inc. The remaining authors disclose no competing interests.

## Additional information

**Correspondence and requests for materials** should be addressed to Daniel J. Weiner or Elise B. Robinson.

# Reporting Summary

## Statistics

For all statistical analyses, confirm that the following items are present in the figure legend, table legend, main text, or Methods section.

| n/a | Confirmed | |
|---|---|---|
| ☐ | ☒ | The exact sample size (*n*) for each experimental group/condition, given as a discrete number and unit of measurement |
| ☒ | ☐ | A statement on whether measurements were taken from distinct samples or whether the same sample was measured repeatedly |
| ☐ | ☒ | The statistical test(s) used AND whether they are one- or two-sided <br> *Only common tests should be described solely by name; describe more complex techniques in the Methods section.* |
| ☐ | ☒ | A description of all covariates tested |
| ☐ | ☒ | A description of any assumptions or corrections, such as tests of normality and adjustment for multiple comparisons |
| ☐ | ☒ | A full description of the statistical parameters including central tendency (e.g. means) or other basic estimates (e.g. regression coefficient) AND variation (e.g. standard deviation) or associated estimates of uncertainty (e.g. confidence intervals) |
| ☐ | ☒ | For null hypothesis testing, the test statistic (e.g. *F*, *t*, *r*) with confidence intervals, effect sizes, degrees of freedom and *P* value noted <br> *Give P values as exact values whenever suitable.* |
| ☒ | ☐ | For Bayesian analysis, information on the choice of priors and Markov chain Monte Carlo settings |
| ☒ | ☐ | For hierarchical and complex designs, identification of the appropriate level for tests and full reporting of outcomes |
| ☐ | ☒ | Estimates of effect sizes (e.g. Cohen's *d*, Pearson's *r*), indicating how they were calculated |

*Our web collection on statistics for biologists contains articles on many of the points above.*

## Software and code

Policy information about availability of computer code

| | |
|---|---|
| Data collection | No data was collected for primary use in this manuscript |
| Data analysis | Data was analyzed using R v4.1 and Python v3.7 |

For manuscripts utilizing custom algorithms or software that are central to the research but not yet described in published literature, software must be made available to editors and reviewers. We strongly encourage code deposition in a community repository (e.g. GitHub). See the Nature Portfolio guidelines for submitting code & software for further information.

## Data

Policy information about availability of data

All manuscripts must include a data availability statement. This statement should provide the following information, where applicable:
- Accession codes, unique identifiers, or web links for publicly available datasets
- A description of any restrictions on data availability
- For clinical datasets or third party data, please ensure that the statement adheres to our policy

Individual-level genotypes are available via request to the Simons Foundation Autism Research Initiative (sfari.org) and from the Psychiatric Genomics Consortium (https://pgc.unc.edu/) and its contributing data holders. GWAS summary statistics from iPSYCH are available by request from the members of the consortium. RNA-sequencing data is available from repository 2304 at the National Database for Autism Research and the CommonMind Consortium (https://www.nimhgenetics.org/resources/commonmind). Hi-C data is available via the respective referenced publications. GTEx specific expression data is available from the Price Lab repository

## Human research participants

Policy information about <u>studies involving human research participants and Sex and Gender in Research.</u>

| | |
|---|---|
| Reporting on sex and gender | Sex and gender-based analyses were not performed. |
| Population characteristics | Characteristics are available in Supplementary Table 3 |
| Recruitment | No participants were specifically recruited for this study. |
| Ethics oversight | We confirm that this study was reviewed and approved by Partners Human Research of Partners HealthCare. The study name is Molecular Study of Cognitive and Behavioral Variation (IRB: 2015P002376). The Principal Investigator is Elise Robinson. The iPSYCH study was approved by the Danish Data Protection Agency and the Scientific Ethics Committee in Denmark. |

Note that full information on the approval of the study protocol must also be provided in the manuscript.

# Field-specific reporting

Please select the one below that is the best fit for your research. If you are not sure, read the appropriate sections before making your selection.

☒ Life sciences ☐ Behavioural & social sciences ☐ Ecological, evolutionary & environmental sciences

For a reference copy of the document with all sections, see nature.com/documents/nr-reporting-summary-flat.pdf

# Life sciences study design

All studies must disclose on these points even when the disclosure is negative.

| | |
|---|---|
| Sample size | The largest sample sizes from the available data were used throughout the analysis. |
| Data exclusions | No data were excluded from analysis. |
| Replication | All experiments were replicated when independent data was available, including: S-pTDT identification of 16p (SSC+SPARK; PGC), regional PGS association to expression (single-nucleus RNA-seq data; bulk RNA-seq data); chromatin contact analysis (LCL; cortical plate). No replication attempts failed. |
| Randomization | Randomization was not relevant to this study since no interventions were performed. |
| Blinding | Blinding was not relevant to this study since no interventions were performed. |

# Reporting for specific materials, systems and methods

We require information from authors about some types of materials, experimental systems and methods used in many studies. Here, indicate whether each material, system or method listed is relevant to your study. If you are not sure if a list item applies to your research, read the appropriate section before selecting a response.

### Materials & experimental systems

| n/a | Involved in the study |
|---|---|
| ☒ | Antibodies |
| ☒ | Eukaryotic cell lines |
| ☒ | Palaeontology and archaeology |
| ☒ | Animals and other organisms |
| ☒ | Clinical data |
| ☒ | Dual use research of concern |

### Methods

| n/a | Involved in the study |
|---|---|
| ☒ | ChIP-seq |
| ☒ | Flow cytometry |
| ☒ | MRI-based neuroimaging |

