## [Peer Review File · Nature Genetics]

Peer Review Information

Manuscript Title: title

Corresponding author name(s): Daniel Weiner

Reviewer Comments & Decisions:

Decision Letter, initial version:

17th June 2022

Dear Dan,

Your Article "Statistical and functional convergence of common and rare variant risk for autism spectrum disorders at chromosome 16p" has been seen by two referees. You will see from their comments below that, while they find your work of interest, they have raised several relevant points. We are interested in the possibility of publishing your study in Nature Genetics, but we would like to consider your response to these points in the form of a revised manuscript before we make a final decision on publication.

To guide the scope of the revisions, the editors discuss the referee reports in detail within the team, including with the chief editor, with a view to identifying key priorities that should be addressed in revision, and sometimes overruling referee requests that are deemed beyond the scope of the current study. In this case, we ask that you address all technical queries with suitable clarifications and revisions and that you perform additional in silico analyses to explore the functional roles of dysregulated genes on 16p, as requested by Reviewer #2. If feasible, we also encourage you to assess the transcriptional consequences of the 16p11.2 distal deletion, as suggested by Reviewer #1. We hope you will find this prioritized set of referee points to be useful when revising your study. Please do not hesitate to get in touch if you would like to discuss these issues further.

We therefore invite you to revise your manuscript taking into account all reviewer and editor comments. Please highlight all changes in the manuscript text file. At this stage we will need you to upload a copy of the manuscript in MS Word .docx or similar editable format.

*1) Include a “Response to referees” document detailing, point-by-point, how you addressed each referee comment. If no action was taken to address a point, you must provide a compelling argument. This response will be sent back to the referees along with the revised manuscript.

*2) If you have not done so already please begin to revise your manuscript so that it conforms to our Article format instructions, available [here](http://www.nature.com/ng/authors/article_types/index.html). Refer also to any guidelines provided in this letter.

[REDACTED]

We hope to receive your revised manuscript within 8-12 weeks. If you cannot send it within this time, please let us know.

Please do not hesitate to contact me if you have any questions or would like to discuss these revisions

further.

Sincerely,
Kyle

Kyle Vogan, PhD
Senior Editor
Nature Genetics
<https://orcid.org/0000-0001-9565-9665>

Referee expertise:

Referee #1: Genetics, structural variation, neuropsychiatric disorders

Referee #2: Genetics, multi-omics, computational methods

Referee #3: Genetics, structural variation, neurodevelopmental disorders

Reviewers' Comments:

Reviewer #1:

3Remarks to the Author:

Authors introduce a stratified-pTDT (S-pTDT), which estimates transmission in parent-child trios of PRS constructed from regions/blocks of the genome. Authors tested whether S-pTDT could identify any regions of the genome with transmission of ASD polygenic risk significantly over or under genome-wide expectation for large blocks of the genome. Authors estimated transmission two large trio samples and the transmission of regional polygenic risk for ASD is correlated between the two samples. Partitions with large S-pTDT z-scores cluster on 16p (0-33Mb).

The 16p does not contain a genome-wide significant locus for ASD, authors still sought to determine whether the S-pTDT signal at 16p could be explained by one or a small number of common variant associations - single driving locus in the region was not found and authors verified that the over-transmission of ASD polygenic risk at 16p is not driven by CNV carriers in their data.

16p is gene rich and many of the genes are only expressed in the brain. Gene density, density of brain-specific genes or density of constrained genes, based on authors' inquiries, cannot explain the region's degree of polygenic over-transmission.

Interestingly, authors observe across independent cohorts that increased 16p ASD PRS is associated with an average decrease in expression of brain expressed genes within the 16p region. Also, in vitro deletion of the 16p11.2 locus was associated with decreased average expression of 200 neuronally expressed genes on chromosome 16p. Furthermore, authors suggest that the 16p region may have increased within-region chromatin contact, which could explain the apparent non-independence of genetic and expression variation at mega-base scale. Authors hypothesize that this diffusely elevated within-region contact at 16p could facilitate the influence of regional polygenic effects on gene expression across 16p, via complex distal regulatory interactions. Lastly authors conclude that the 16p11.2 CNV has increased physical interaction with the telomeric region and the 3D conformation of 16p may mediate convergent ASD-related genetic effects on gene expression via regulatory interactions across mega-bases of separation. Based on these observations authors present the "Integrative model of ASD liability at 16p".

Comments

1. The samples studied are large and impressive, the analyses are transparent, novel and sound and the model is interesting. The main weakness is how modest the mean expression effects are for the 16p region, particularly when compared to the decrease in gene expression associated with heterozygous gene deletion (16p11.2 CNV).

2. The deletion described and studied in this manuscript is the 16p11.2 proximal deletion explaining up to 1% of autistic cases. It is particularly interesting that distal to this locus is another recurrent deletion, 16p11.2 distal deletion, not mentioned in the manuscript. That deletion, the 16p11.2 distal deletion confers high-risk of very similar phenotypes (including autism, cognitive impairments and obesity). If that deletion, the 16p11.2 distal deletion, also affects gene expression on 16p in similar manner as the 16p11.2 proximal deletion and the 16p ASD PRS, the story would be more convincing. Thus, my recommendation is to include as well data on the 16p11.2 distal deletion in the manuscript.

References for the distal deletion:

a. Dose response of the 16p11.2 distal copy number variant on intracranial volume and basal ganglia. Ida E Sønderby et al. *Molecular Psychiatry* (2020) 25:584–602 <https://doi.org/10.1038/s41380-018-0118-1>

Reviewer #2:

None

Reviewer #3:

Remarks to the Author:

In this study, Weiner et al. investigate the feasibility of extracting biological insight from a large genomic region and understanding how it is associated with risk for autism spectrum disorder. They identified the 33 Mb short arm of chromosome 16 as harboring the greatest excess of common polygenic risk for ASD. Analysis of bulk and single-cell RNA-sequencing data from post-mortem human brain samples revealed that common polygenic risk for ASD within 16p is associated with decreased average expression of genes throughout this 33-Mb region.

They subsequently use isogenic neuronal cell lines with CRISPR/Cas9-mediated deletion of 16p11.2 to show that the deletion is also associated with depressed average gene expression across the short arm of chromosome 16. The effects of the rare deletion and diffuse common variation were correlated at the level of individual genes. Their results also suggest that very dense 3D chromatin contact within the short arm of chromosome 16 may coordinate genetic and transcriptional disease liability across this region.

5This study focuses on a large genomic segment rather than on specific genes. It is original and advances the field. The claims are supported by the data. In particular, the authors provide a rather convincing link between the transcriptional effects of rare and common variants implicated in ASD.

My major comment is that there is a lack of functional characterization of this large group of genes on the short arm of 16p. If these genes are co-regulated, this would imply that they are implicated in shared functional modules. The study would benefit from a functional characterization: i.e., Are genes within this genomic block enriched in well-known functional modules? Or in modules identified by contrasting gene expression in the brains of individuals with ASD and controls?

Specific comments:

In the 2nd paragraph of the results:

"... we constructed stratified PRS from adjacent blocks of SNPs, yielding 2,006 (often overlapping) partitions collectively covering the whole genome (median number of SNPs per block: 3,000, minimum length: 4.3Mb, maximum length: 52.9Mb, median length: 11.7Mb, Supplementary Figure 3, Methods)..." It is unclear how they defined these genomic blocks of very different sizes. Was it based on the number of LD blocks, the number of genes? Or completely random?

Figure 1E. The number of blocks removed stops at $n=25$. Is that just because there was no more effect? One would expect that the SE of the pTDT would get larger as more blocks are removed, but that doesn't appear to be the case. It also seems like there is a trend that may become significantly protective at one point. What would happen beyond $n=25$? In other words, once authors remove the most over transmitted blocks, are there protective blocks in the 16p11.2 short arm?

As a sensitivity analysis, the authors performed an analogous analysis using a cohort of ADHD trios and an external ADHD GWAS and they did not replicate the finding in ADHD. One could argue that ADHD may be the worst condition to perform such a sensitivity analysis since the PRS doesn't explain much variance. Schizophrenia would appear to be much more relevant. The PRS is more robust, cohorts are larger, and the 16p11.2 locus is associated with schizophrenia.

The relationship between gene density and over-transmission is an important point and should be represented in a figure in the main text.

The authors asked whether, on average, the 200 neuronally expressed genes on 16p were differentially

expressed in response to the 16p11.2 deletion. Genes on 16p had significantly lower expression in the deletion lines. The deletion's effect on 16p genes differed from the effect on all other 8,533 neuronally expressed genes in the genome ($P = 0.02$ - somewhat of a trend -), whose expression was not, on average, changed by the deletion ($P = 0.43$).

Can the authors provide information on the non-neuronally expressed genes on the short arm? Would these represent a better "control group", providing a stronger contrast?
Shouldn't the Y-axis of figure 2C represent "fold change" instead of t-stat?

Increased ASD PRS within 16p was associated with decreased expression in glutamatergic neurons of genes through the 16p region. Do the authors observe an increase in the variance of gene expression? In other words, is this a simple shift in mean expression, or do results also suggest that there may also be some genes with an increase in expression?

Authors show that the CNV-telomeric contacts ($n = 291$ 100kb x 100kb contacts) are 2.9x more frequent than contacts between distance-matched control regions on 16p ($n = 1,808$ 100kb x 100kb contacts, $P < 1e-10$).

However, the 16p11.2 region is 30MB away from the telomeric region. I don't see, therefore how they can test distance-matched regions on the short arm. Distance-matched regions would only be found downstream of the 16p11.2 locus on the long arm.

Authors suggest that the entire short arm may represent a group of co-regulated genes involved in ASD. It would be necessary to demonstrate that this is the case and test the enrichment of these genes in known functional modules (in health and disease. i.e.). For example, are these gene enriched differentially expressed genes obtained by contrasting gene expression in the brains of individuals with ASD and controls. This data is available, and the analysis should be straightforward.

Author Rebuttal to Initial comments

Response to reviewers

We thank the reviewers for their thoughtful comments and have responded point-by-point below. We have also highlighted any corresponding changes in the main text.

Reviewer #1:

Comment #1.1: *“The deletion described and studied in this manuscript is the 16p11.2 proximal deletion explaining up to 1% of autistic cases. It is particularly interesting that distal to this locus is another recurrent deletion, 16p11.2 distal deletion, not mentioned in the manuscript. That deletion, the 16p11.2 distal deletion confers high-risk of very similar phenotypes (including autism, cognitive impairments and obesity). If that deletion, the 16p11.2 distal deletion, also affects gene expression on 16p in similar manner as the 16p11.2 proximal deletion and the 16p ASD PRS, the story would be more convincing. Thus, my recommendation is to include as well data on the 16p11.2 distal deletion in the manuscript.”*

We thank the reviewer for this thoughtful comment about the possibility that other ASD-associated CNVs – especially those located on 16p – may confer disease liability through similar mechanisms to the proximal 16p11.2 studied here. The reviewer notes the potential relevance of the 16p11.2 distal deletion, and we agree this proximate and disease-associated CNV would be interesting to investigate. However, as far as we can tell, there are no published whole-genome RNA-sequencing datasets of the 16p11.2 distal deletion, including in the provided reference of Sønderby et al. It is also materially infeasible for us to generate additional isogenic distal deletions at this time. Without RNA-sequencing, we are unable to evaluate our model for this deletion.

That said, we are very interested in whether other neuropsychiatric CNVs confer disease liability through similar mechanisms to those described for the proximal 16p11.2 CNV. We are actively testing this hypothesis in our research group, and we look forward to sharing our findings with the community in the future. Finally, we clarified in the manuscript that we analyzed the proximal and not the distal deletion (page 3).

Reviewer #3:

Comment #3.1: *“My major comment is that there is a lack of functional characterization of this large group of genes on the short arm of 16p. If these genes are co-regulated, this would imply that they are implicated in shared functional modules. The study would benefit from a functional characterization: i.e., Are genes within this genomic block enriched in well-known functional modules? Or in modules identified by contrasting gene expression in the brains of individuals with ASD and controls?”*

We thank the reviewer for raising this important question – we are also extremely eager to understand the aggregated downstream functional consequence of genetic variation in the region.

We first performed gene ontology (GO) analysis to evaluate enrichment of genes on 16p in annotated biological pathways (<http://geneontology.org/>). We used the same 17,909 genes from the gene density analysis as reference genes. We tested for enrichment of all genes on 16p (midpoint < 32,000,000 bp, n = 432 genes) across three classes of annotations: biological process, molecular function, and cellular component.

The GO analysis for molecular function and cellular component returned multiple bonferroni-significant enrichments: multiple lipid/fatty acid pathways (Fatty-acyl-CoA synthase activity, Butyrate-CoA ligase activity,

Medium-chain fatty acid-CoA ligase activity, >20x enrichment for each), and hemoglobin complex (19x enrichment). The lipid/fatty acid pathways return an enrichment because there are 5 acyl-CoA-synthase genes located within 500kb of each other on 16p around Mb 20. Similarly, the hemoglobin complex pathway returns an enrichment because 4 hemoglobin subunits are clustered together within 100kb of each other at the start of chromosome 16. These examples raise a critical point: since functionally similar genes are often clustered together in the genome (Andrews et al. 2015 Genome Research), a gene set enrichment signal will be dominated by whichever functional cluster of genes happens to be located within the region of interest. Thus, we do not believe that canonical gene set enrichment approaches are suited to regional enrichment analysis. That said, it is also possible that decreased expression across 16p does not exert direct phenotypic effect, but instead propagates to interact with gene/protein networks elsewhere in the cell or cellular network. As cell-type specific interaction networks come on line in coming years, we look forward to integrating with our analyses.

Next, we tested the hypothesis that genes on 16p are over-represented in analysis of differential expression in the brains of individuals with ASD vs. controls. We identified differentially expressed genes between ASD cases (n = 51) and controls (n = 936) from a recent publication and retained those significantly variable at a bonferroni-significant level (n = 83 genes) (Gandal et al 2018 Science). We used a chi-squared test for over-representation of genes on 16p (n = 383 in Gandal dataset) in this n = 83 differentially expressed gene set. We did not find over- or under-representation of 16p-related genes in the Gandal DEG set ($p > 0.05$). Given the genetic heterogeneity of ASD, among the other non-genetic factors contributing to expression variability between ASD cases and controls, we do not find it surprising that there is no overlap between these gene sets.

In summary, we believe it is most likely that the genes on 16p – modulated by both common variants on 16p and the 16p11.2 deletion – are integrated in a complex network that is not ascertainable through canonical gene set enrichment approaches. We are engaging with members of the community to develop approaches to extract additional biological meaning out of regional variation in gene expression. We have added these analyses to the main text (page 6) and supplement (Supplementary Table 1).

Comment #3.2: *"In the 2nd paragraph of the results: "... we constructed stratified PRS from adjacent blocks of SNPs, yielding 2,006 (often overlapping) partitions collectively covering the whole genome (median number of SNPs per block: 3,000, minimum length: 4.3Mb, maximum length: 52.9Mb, median length: 11.7Mb, Supplementary Figure 3, Methods)..." It is unclear how they defined these genomic blocks of very different sizes. Was it based on the number of LD blocks, the number of genes? Or completely random?"*

We agree this important section of the methods deserves additional detail in the text; we have expanded this methods section in the manuscript (page 18). In brief, for creating genomic blocks of 2,000 SNPs, we identified the first PRS SNP on chromosome 1 (the SNP closest to the first base pair), counted 2,000 PRS SNP, and called that the first partition. Then, we counted the next 2,000 PRS SNPs on chromosome 1, called that the next partition, etc, until we ran out of SNPs on chromosome 1. Then we started the same process on chromosome 2, etc. We repeated this for blocks of different sizes (3,000 SNPs, 4,000 SNPs, 5,000 SNPs, and 6,000 SNPs), as well as repeated the entire process starting at the ends of chromosomes and going backwards. The partitions were not based on LD blocks, nor on genes/gene density.

Comment #3.3: *"Figure 1E. The number of blocks removed stops at n=25. Is that just because there was no more effect?"*

In Figure 1E, the number of blocks removed stops at n = 25 because that is the number of these blocks located within 16p (median length of each block: 1.31 Mb). We have clarified this point in the text (page 19).

Comment #3.4: “One would expect that the SE of the pTDT would get larger as more blocks are removed, but that doesn't appear to be the case.”

The SE of the S-pTDT decreases with larger sample size (right plot) but does not vary with the number of SNPs in the PRS partition (left plot).

Comment #3.5: (Figure 1E) “It also seems like there is a trend that may become significantly protective at one point. What would happen beyond =25? In other words, once authors remove the most over transmitted blocks, are there protective blocks in the 16p11.2 short arm?”

Some of the blocks on 16p are (non-significantly) under-transmitted to ASD probands (Supplementary Figure 8), which indeed reflects a trend towards the common variants in that block being protective. This is not unique to 16p, but reflects a genome-wide pattern, where many regions of ASD common variation are under-transmitted in our three trio cohorts (Supplementary Figure 4). This reflects the genetic variability among our trio cohorts, where it is only with some probability at a given locus that an ASD proband inherited the liability-increasing haplotype.

Comment #3.6: “As a sensitivity analysis, the authors performed an analogous analysis using a cohort of ADHD trios and an external ADHD GWAS and they did not replicate the finding in ADHD. One could argue that ADHD may be the worst condition to perform such a sensitivity analysis since the PRS doesn't explain much variance. Schizophrenia would appear to be much more relevant. The PRS is more robust, cohorts are larger, and the 16p11.2 locus is associated with schizophrenia.”

We agree that the SCZ PRS is a more predictive instrument for SCZ than is the ADHD PRS for ADHD. However, relative to the ASD PRS, the ADHD PRS performs well (ADHD Nagelkerke's $R^2 = 5.5\%$, Demontis et al. 2019 Nature Genetics, vs. ASD Nagelkerke's $R^2 = 2.5\%$, Grove et al. 2019 Nature Genetics). Regarding cohorts, the schizophrenia cohorts in the PGC are case-control design and not trio, which is required for the within-family transmission analysis of S-pTDT. In contrast, we were able to use ADHD trios from the PGC. Finally, the 16p11.2 locus is also associated with ADHD (Niarchou et al. 2019 Translational Psychiatry).

Comment #3.7: “The relationship between gene density and over-transmission is an important point and should be represented in a figure in the main text.”

We agree this is an important point and have moved one of the panels relating gene-density and over-transmission to the main text as an inset to Figure 1F.

Comment #3.8: “The authors asked whether, on average, the 200 neuronally expressed genes on 16p were differentially expressed in response to the 16p11.2 deletion. Genes on 16p had significantly lower expression in the deletion lines. The deletion’s effect on 16p genes differed from the effect on all other 8,533 neuronally expressed genes in the genome ($P = 0.02$ - somewhat of a trend -), whose expression was not, on average, changed by the deletion ($P = 0.43$). Can the authors provide information on the non-neuronally expressed genes on the short arm? Would these represent a better “control group”, providing a stronger contrast? Shouldn’t the Y-axis of figure 2C represent “fold change” instead of t-stat?”

We thank the reviewer for this thoughtful comment, and agree that the low expression condition should be included in the analysis for comparison. We have assessed the effect in low expression genes in both the isogenic 16p11.2 deletion lines, and using the regional PRS, and confirmed that the effect on decreased expression is attenuated in those lower expressed genes in both sets of analyses. We have summarized the findings in a figure below in this response. In the main text and supplement, we have added text describing the analyses and results for both the isogenic deletion (main text page 9, supplementary figure 16) and for the regional PRS approach (main text page 11, supplementary figure 20).

Regarding units in Figure 2C, the left panel is in log(fold-change) for intuitive interpretability, while the right panel is the statistical comparison of the two groups that incorporates uncertainty in the fold-change estimates, hence displaying the changes in uncertainty-normalized t-statistics.

Comment #3.9: “Increased ASD PRS within 16p was associated with decreased expression in glutamatergic neurons of genes through the 16p region. Do the authors observe an increase in the variance of gene expression? In other words, is this a simple shift in mean expression, or do results also suggest that there may also be some genes with an increase in expression?”

Thank you for this interesting question. To explore this further, for each 33Mb region of the genome (“partition”), we associated regional PRS with expression of each gene and extracted the association t-statistic across our 544 samples. There was no association between either the partition’s mean(t-statistic) or |mean(t-statistic)| and variance(t-statistic) across partitions (see plots below, where each dot is a partition with 16p in blue. $P > 0.05$ for each). For 16p specifically, we do not see a dramatic increase in expression variance given the decrease in expression averaged across all genes.

Comment #3.10: “Authors show that the CNV-telomeric contacts ($n = 291$ 100kb x 100kb contacts) are 2.9x more frequent than contacts between distance-matched control regions on 16p ($n = 1,808$ 100kb x 100kb contacts, $P < 1e-10$). However, the 16p11.2 region is 30MB away from the telomeric region. I don’t see, therefore how they can test distance-matched regions on the short arm. Distance-matched regions would only be found downstream of the 16p11.2 locus on the long arm.”

We define the telomeric region from 0 Mb to 5.2 Mb on chromosome 16, while the 16p11.2 (proximal) CNV ranges from 29.5Mb-30.2Mb. The contacts between these regions are denoted in Figure 4C in the blue shaded rectangle inside the larger triangular contact matrix. The range of distances encompassed between these contacts begins at 24.3Mb in distance (contact between Mb 5.2 of the telomeric region and Mb 29.5 of the CNV: $29.5 - 5.2 = 24.3$) and extends to 30.2Mb in distance (contact between 30.2Mb of the CNV to 0 Mb of the telomeric region). Therefore, the distance-matched control regions are contacts on 16p that span 24.3Mb to 30.2Mb. There are many such 100kb x 100kb contacts ($n = 1,808$), and such contacts are denoted in the red shaded trapezoid in Figure 4C (for example, contact between Mb 6 and Mb 31 = 25 Mb apart).

Decision Letter, first revision:

Our ref: NG-A59672R

4th August 2022

Dear Dan,

Your revised manuscript "Statistical and functional convergence of common and rare genetic influences on autism at chromosome 16p" (NG-A59672R) has been seen by the original referees. As you will see from their comments below, they find that the paper has improved in revision, and therefore we will be happy in principle to publish it in Nature Genetics as an Article pending final revisions to address the referees' remaining points and to comply with our editorial and formatting guidelines.

We are now performing detailed checks on your paper and we will send you a checklist detailing our editorial and formatting requirements soon. Please do not upload the final materials or make any revisions until you receive this additional information from us.

Thank you again for your interest in Nature Genetics. Please do not hesitate to contact me if you have any questions.

Sincerely,
Kyle

Kyle Vogan, PhD
Senior Editor
Nature Genetics
<https://orcid.org/0000-0001-9565-9665>

Reviewer #1 (Remarks to the Author):

Reviewer #1: Comment #1.1: “The deletion described and studied in this manuscript is the 16p11.2 proximal deletion explaining up to 1% of autistic cases. It is particularly interesting that distal to this locus is another recurrent deletion, 16p11.2 distal deletion, not mentioned in the manuscript. That deletion, the 16p11.2 distal deletion confers high-risk of very similar phenotypes (including autism, cognitive impairments and obesity). If that deletion, the 16p11.2 distal deletion, also affects gene expression on 16p in similar manner as the 16p11.2 proximal deletion and the 16p ASD PRS, the story would be more convincing. Thus, my recommendation is to include as well data on the 16p11.2 distal deletion in the manuscript.”

Authors reply

We thank the reviewer for this thoughtful comment about the possibility that other ASD-associated CNVs – especially those located on 16p – may confer disease liability through similar mechanisms to the proximal 16p11.2 studied here. The reviewer notes the potential relevance of the 16p11.2 distal deletion, and we agree this proximate and disease-associated CNV would be interesting to investigate. However, as far as we can tell, there are no published whole-genome RNA-sequencing datasets of the 16p11.2 distal deletion, including in the provided reference of Søndersby et al. It is also materially infeasible for us to generate additional isogenic distal deletions at this time. Without RNA-sequencing, we are unable to evaluate our model for this deletion. That said, we are very interested in whether other neuropsychiatric CNVs confer disease liability through similar mechanisms to those described for the proximal 16p11.2 CNV. We are actively testing this hypothesis in our research group, and we look forward to sharing our findings with the community in the future. Finally, we clarified in the manuscript that we analyzed the proximal and not the distal deletion (page 3).

Further comments from Reviewer #1:

I find the results presented in this manuscript most interesting. However, they should be confirmed. Authors can identify RNA-sequenced samples suitable for confirming their findings or they can analyze the isogenic neuronal cell lines with CRISPR/Cas9-mediated “distal” deletion of 16p11.2. That may reveal that the deletion also associates with depressed average gene expression across 16p and, hence, confirm the findings.

Reviewer #3 (Remarks to the Author):

The authors responded to all comments and questions in a satisfactory way.

The only response that remains unclear relates to comment 3.10.

The authors describe the contacts between the 16p11.2 region and the telomeric region ranging from 24.3 to 30.2 Mb in distance.

They give an example of the distance-matched control regions between MB6 and MB31. However, this contact is beyond the 16p11.2 region. Does this mean that the distance-matched control contacts do not include the 16p11.2 region?

Author Rebuttal to Initial comments

Dear Kyle,

Thank you for sharing the reviewer comments. Please see our responses below:

Response to reviewer #1

We are glad the reviewer finds our manuscript of great interest. We interpret the reviewer's specific request here as asking for analysis of an additional isogenic CRISPR-generated lines of either the proximal or distal 16p11.2 deletion. While we've reviewed the literature and inquired broadly, such a resource does not seem to currently exist, unfortunately.

We're happy to reflect more on the one result in question (the data in Figure 2). We do find support for the observation from other analyses presented in the manuscript, including a) convergence with expression effects to the 16p ASD PRS (Figure 4A), b) elevated chromatin contact between the 16p11.2 deletion region and the telomeric region of convergent effect (Figure 4C), and c) lack of a similar observation at the 15q locus supporting the specificity of the 16p11.2 deletion effect on regional gene expression (Supplementary Figure 17). We will also add a note to the discussion that replication using further isogenic lines or very large patient-derived samples will be valuable once those resources are developed.

Should the NG editors or the reviewer have other questions or suggestions, we're happy to discuss.

Response to reviewer #3

10We are glad the reviewer finds our responses satisfactory. With regards to comment 3.10: yes, that is correct that almost all of the control contacts do not include the 0.7Mb 16p11.2 region. Three-dimensional contact frequencies as assayed by Hi-C are strongly dependent on the distance between the contact loci (decaying with distance). Thus, we defined control regions based on their distance in such a way that the range of control contact distances (red trapezoid in Figure 4C) is the same as the range of contact distances between the 16p11.2 region and the telomeric contact region (blue rectangle in Figure 4C). Hopefully this is clarifying.

Final Decision Letter:

In reply please quote: NG-A59672R1 Weiner

15th September 2022

Dear Dan,

I am delighted to say that your manuscript "Statistical and functional convergence of common and rare genetic influences on autism at chromosome 16p" has been accepted for publication in an upcoming issue of Nature Genetics.

Due to the importance of these deadlines, we ask that you please let us know now whether you will be difficult to contact over the next month. If this is the case, we ask you provide us with the contact

11information (email, phone and fax) of someone who will be able to check the proofs on your behalf, and who will be available to address any last-minute problems.

Your paper will be published online after we receive your corrections and will appear in print in the next available issue. You can find out your date of online publication by contacting the Nature Press Office (press@nature.com) after sending your e-proof corrections. Now is the time to inform your Public Relations or Press Office about your paper, as they might be interested in promoting its publication. This will allow them time to prepare an accurate and satisfactory press release. Include your manuscript tracking number (NG-A59672R1) and the name of the journal, which they will need when they contact our Press Office.

Before your paper is published online, we will be distributing a press release to news organizations worldwide, which may very well include details of your work. We are happy for your institution or funding agency to prepare its own press release, but it must mention the embargo date and Nature Genetics. Our Press Office may contact you closer to the time of publication, but if you or your Press Office have any enquiries in the meantime, please contact press@nature.com.

Please note that Nature Genetics is a Transformative Journal (TJ). Authors may publish their research with us through the traditional subscription access route or make their paper immediately open access through payment of an article-processing charge (APC). Authors will not be required to make a final decision about access to their article until it has been accepted. [Find out more about Transformative Journals](https://www.springernature.com/gp/open-research/transformative-journals)

Authors may need to take specific actions to achieve [compliance](https://www.springernature.com/gp/open-research/funding/policy-compliance-faqs) with funder and institutional open access mandates. If your research is supported by a funder that requires immediate open access (e.g. according to [Plan S principles](https://www.springernature.com/gp/open-research/plan-s-compliance)), then you should select the gold OA route, and we will direct you to the compliant route where possible. For authors selecting the subscription publication route, the journal's standard licensing terms will need

to be accepted, including <https://www.nature.com/nature-portfolio/editorial-policies/self-archiving-and-license-to-publish>. Those licensing terms will supersede any other terms that the author or any third party may assert apply to any version of the manuscript.

Please note that Nature Portfolio offers an immediate open access option only for papers that were first submitted after 1 January 2021.

If you have not already done so, we invite you to upload the step-by-step protocols used in this manuscript to the Protocols Exchange, part of our on-line web resource, natureprotocols.com. If you complete the upload by the time you receive your manuscript proofs, we can insert links in your article that lead directly to the protocol details. Your protocol will be made freely available upon publication of your paper. By participating in natureprotocols.com, you are enabling researchers to more readily

reproduce or adapt the methodology you use. Natureprotocols.com is fully searchable, providing your protocols and paper with increased utility and visibility. Please submit your protocol to <https://protocolexchange.researchsquare.com/>. After entering your nature.com username and password you will need to enter your manuscript number (NG-A59672R1). Further information can be found at <https://www.nature.com/nature-portfolio/editorial-policies/reporting-standards#protocols>

Sincerely,
Kyle

Kyle Vogan, PhD
Senior Editor
Nature Genetics
<https://orcid.org/0000-0001-9565-9665>